# Thermal Inkjet Printing: Prospects and Applications in the Development of Medicine

**Md Jasim Uddin** [1,2,3], **Jasmin Hassan** [3] and **Dennis Douroumis** [1,2,*]

1    Faculty of Engineering and Science, University of Greenwich at Medway, Chatham Maritime, Chatham, Kent ME4 4TB, UK

2    Center for Innovation, Process Engineering & Research, University of Greenwich at Medway, Chatham Maritime, Chatham, Kent ME4 4TB, UK

3    Drug Delivery & Therapeutics Lab, Dhaka 1212, Bangladesh

*    Correspondence: d.douroumis@greenwich.ac.uk

**Abstract:** Over the last 10 years, inkjet printing technologies have advanced significantly and found several applications in the pharmaceutical and biomedical sector. Thermal inkjet printing is one of the most widely used techniques due to its versatility in the development of bioinks for cell printing or biosensors and the potential to fabricate personalized medications of various forms such as films and tablets. In this review, we provide a comprehensive discussion of the principles of inkjet printing technologies highlighting their advantages and limitations. Furthermore, the review covers a wide range of case studies and applications for precision medicine.

**Keywords:** thermal inkjet printing; TIJ; bubble jet printing; personalized treatment; precision medicine

## 1. Introduction

Over the last 20 years, we have encountered a transformation in manufacturing technologies in the area of medicinal products [1–3]. Traditionally marketed medicines are manufactured at fixed doses (one size fits all) targeting a large number of patients in order to reduce the production costs and time to the market. However, the widely varied responses to a particular therapeutic dose in patient populations especially for medicines with narrow therapeutic windows points out the limitations of generalized mass manufacturing [1,4]. Moreover, there is a growing number of patients worldwide with chronic diseases who have to take multiple doses of medicines per day, called polypharmacy, which increases the risk for side effects and drug–disease interactions [5]. Currently, swift advances in gene sequencing technology along with increased knowledge of genomics and better understanding of diseases on molecular level coupled with the use of toxicogenomic markers have opened a door for personalized medicine that will possibly bring a revolution in the conventional treatment approaches as well as in pharmaceutical industry [6–9]. For the materialization of these advances in personalized medicines, a wide range of 2D and 3D printing technologies have been introduced as appropriate for manufacturing print-on-demand medicinal products. Inkjet printing (IJP) technology is considered an ideal approach as it is cost effective [1,10–14] with high precision, repeatability, robustness, and high-throughput (Figure 1). Due to its wide applicability, inkjet printing has been extensively used for pharmaceutical applications and tissue engineering [15–24].

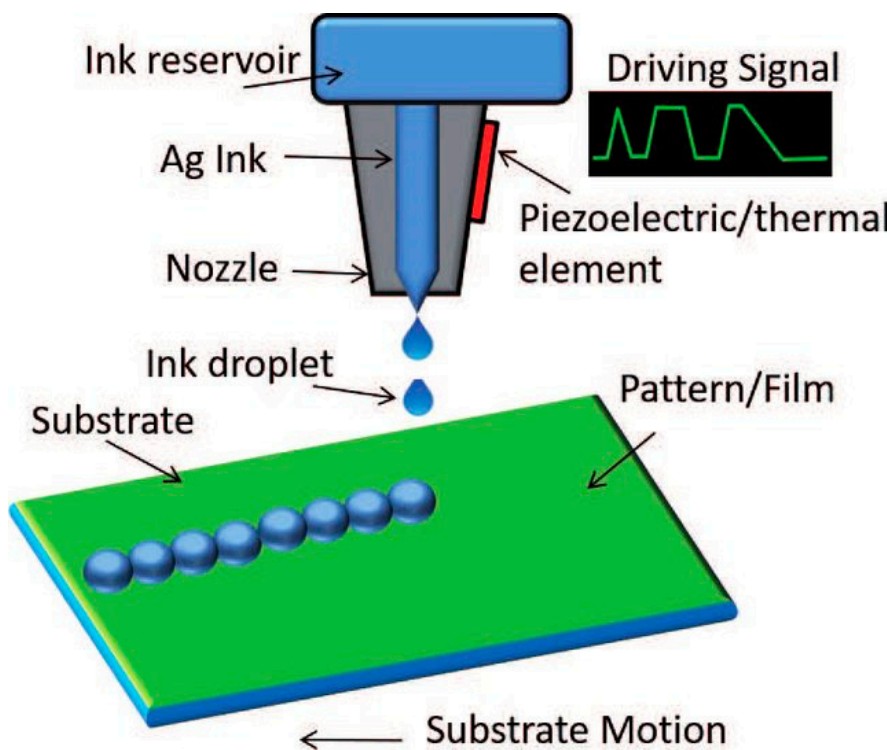

**Figure 1.** Schematic illustration of inkjet printing technology [25].

### 1.1. Inkjet Printing Technology

Inkjet printing is a reprographic method that provides for the controlled deposition of a small drop of ink (e.g., biological, synthetic and any form of therapeutic or nontherapeutic substances) on a substrate [26,27].

Today, this widely known digital printing technology that was originally developed to transfer electronic data on paper is present in almost every office and household as a common technique for printing text or graphics [26–29]. Being a noncontact deposition and direct-patterning technique, it provides minimal contamination and waste of therapeutic sample, respectively [11,30–32]. It has also caught the attention of researchers worldwide because of its drop placement accuracy in a precisely fixed amount (typically in volumes of picolitres, pL) of material that can be dispensed without any prior pattern [11,30,33,34].

This material-conserving patterning technique is usually used for the deposition of liquid phase materials that are technically termed ink and that contain the solute as dissolved or dispersed in a solvent. A piezoelectric inkjet printer uses a piezo-ceramic plate to apply ink droplets in order to regulate the ejection. To avoid unwanted interactions between the inks and the plate, a tiny diaphragm is connected to the piezo-ceramic plate. An electric impulse causes a piezo-ceramic plate to distort, and subsequently, the droplet is ejected from the nozzle as a result of the pressure wave this creates. The piezo-ceramic plate returns to its original shape after the electric pulse is removed, and the ink is replaced. These ejected droplets gravitate towards and settled on the surface of the substrate using the momentum obtained during the motion. Subsequently, droplets dry (see detailed schematic representation in Figure 2) via the evaporation of the solvent [35].

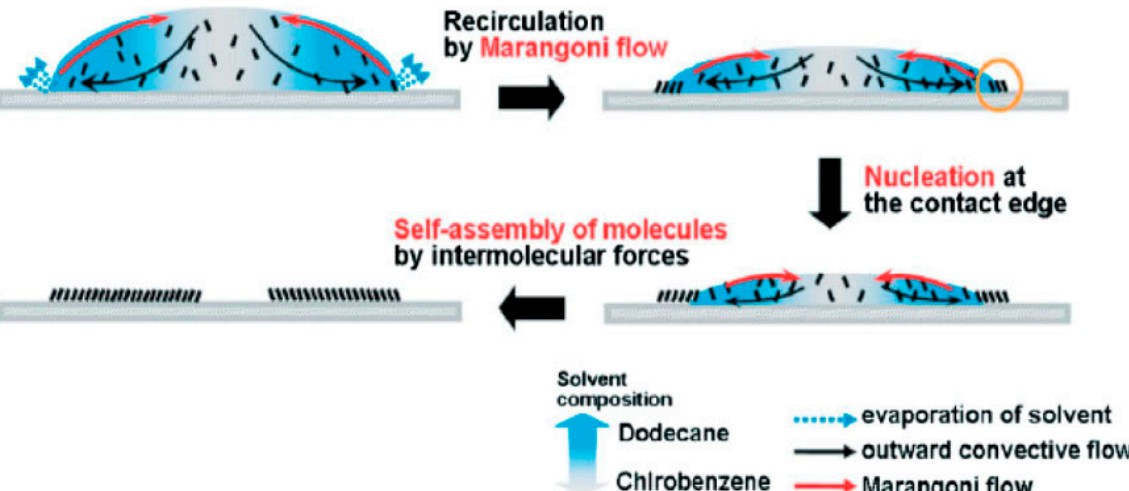

**Figure 2.** Schematic representation of the potential drying process and crystal formation of an organic semiconductor, 6,13-bis((triisopropylsilylethynyl) pentacene (form 25% dodecane) after deposition from a drop-on-demand piezoelectric print head [36].

Before the drying process starts, the droplet first reaches an equilibrium temperature due to the continuous heat loss and evaporation of the solvent into a warmer environment of surroundings at a certain pressure and temperature [37,38]. In the first stage of droplet drying, the drying rate (which is commonly expressed in kg m$^{-2}$ s$^{-1}$) is limited and determined via the energy essential to evaporate the solvent, which leads the heat transport towards the droplet's surface [38,39]. After that, the drying process starts from the surface of the droplets, and the molecules of solvent keep drifting towards the surface from the center, which can be mediated via diffusion allied to the solute, convection of liquid within the droplet or capillary fluid flow [37,38,40]. If the temperature of the surroundings is constant, then the drying rate remains unchanged and determined only by the temperature transfer towards the droplet surface [38,41]. For this reason, the first stage of droplet drying is known as the constant-rate drying stage [38].

The second stage of droplet drying is elucidated by the materials present in droplets. Since the evaporation of liquid occurs in the surface of a droplet, the material concentration increases at the surface. This growing concentration gradient results in diffusional material flux far from the surface and towards the center of the droplet, which is a complex phenomenon [38,39]. Consequently, the diffusional motion of the material towards the droplet's center becomes less than the reduction rate of the droplet diameter because of the constant rate for solvent loss. At this point, crust forms because the higher concentration of materials at the droplet surface leads to a decrease in the drying rate [38,42]. This point is called the locking point or critical point. In the beginning of the second drying stage, a porous solid crust with an internally wet core might be observed in the drying droplet, and drying rate here is now determined by the diffusion or capillary flow rate of the liquid from the wet core via the porous crust. A slowed liquid evaporation still causes the shrinkage of wet core and a considerable increase in the crust towards the droplet's focal point [42,43]. The condensed crust will influence a growing resistance to mass transfer, and therefore a decrease in the drying rate can be observed. Because of that, this second stage is called the falling-rate stage in the droplet-drying process [37]. It infers the presence of lowest possible amount of residual liquid in a single droplet, which can be either an equilibrium amount or residual solvent that cannot be eliminated by drying [38,44]. Hence, the droplet drying rate in the course of time at the falling-rate period might take on different shapes depending on the mechanism and factors of the drying [43,45]. Figure 3 shows a simplified illustration of the two stages of droplet drying.

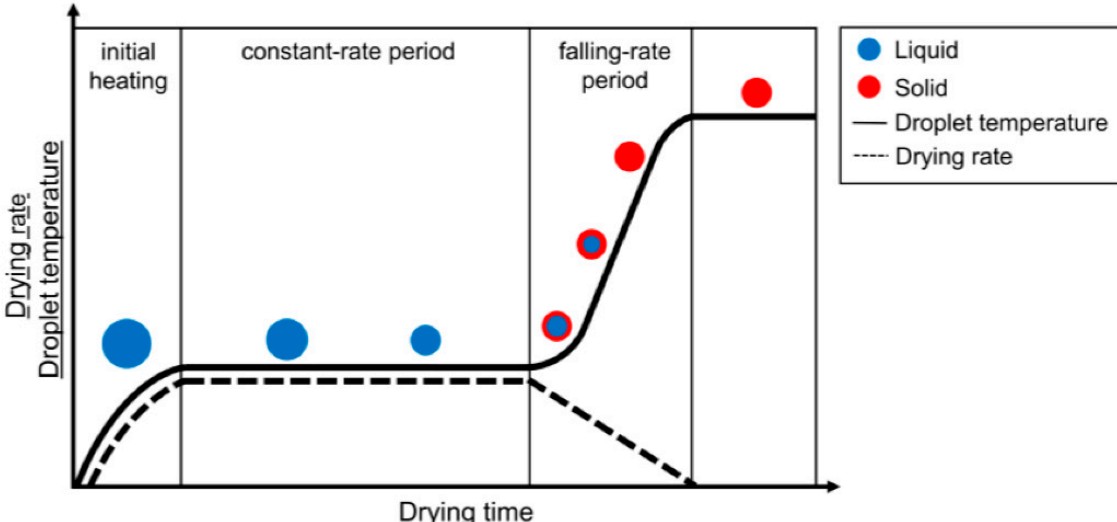

**Figure 3.** Simplified schematic depiction of the different stages a droplet goes through while drying. Drying rate is represented by a dotted line and temperature evolution by a solid line. The solid fraction is shown in red and the liquid fraction in blue [43,45].

In short, the mechanism of inkjet printing comprises three main steps: (1) ejection of ink and droplet formation, (2) liquid–solid interaction after the placement of droplets on the substrate's surface and (3) drying of ink droplet and subsequent solidifying of the printed features to generate a solid deposit [46,47].

Moreover, inkjet printing does not require sophisticated infrastructure such as clean rooms and large-scale facilities [11]. Merely tiny drops are enough to produce superior quality images with higher resolution [48]. Overall, inkjet printing is a better patterning technique in comparison with some of the other available technologies in the market (Table 1) in terms of cost, efficiency, resolution, compatibility with polymer, process, mode of action, flexibility, requirement of environment and material consumption [49,50].

**Table 1.** Comparison of some typical patterning technologies [50].

| SN | Properties | Photolithography | Micro-Contact Printing | Shadow Mask | Inkjet Printing |
|----|-----------|------------------|------------------------|-------------|-----------------|
| 1. | Cost | Extremely high | Medium | Low | Low |
| 2. | Efficiency | Low | High | High | High |
| 3. | Resolution | Extremely high | High | Low | High |
| 4. | Compatibility with polymer | Bad | Bad | Good | Excellent |
| 5. | Process | Multi step | Multi step | Multi step | All in one |
| 6. | Mode of action | Noncontact | Contact | Contact | Noncontact |
| 7. | Flexibility | Bad | Bad | Bad | Good, digital lithography |
| 8. | Requirement of environment | Clean rooms, vibration isolation | Medium | Low | Low |
| 9. | Material consumption | High | Low | Medium | Low |

Inkjet printers present some drawbacks. Mainly, they are expensive because of their two basic requirements: (a) printheads must be well suited to different kinds of inks, for example polymeric or metal-based inks and (b) printing cycles must be executed repeatedly.

Low-cost inkjet printers that are used generally in household and office might be considered for use as simple devices, but they usually cannot perform due to the incompatibility with the bioinks or inks used for laboratorial research purposes. This is especially the case with metal inks due to their viscosity and nozzle occlusion problems. In addition, the printers' multilayer patterned structure makes it quite difficult to print with them [51–54]. An additional issue is the printing of polymers on material surfaces, which leads to adsorbed patterns that are poorly adhesive [55]. Another disadvantage is the coffee ring effect, which causes inkjet-printed insulators to generate a wave-shaped profile where other methods produce perfectly smooth profiles, such as spin-coating [56].

To overcome these issues, researchers have come up with different strategies for both maximizing the benefits that inkjet printing provides and minimizing the disadvantages as well. For example, screen printing provides high-speed printing that is adaptive to commonly available materials and complex multilayer devices. In addition, low-cost inkjet piezoelectric printers provide good spatial resolution (for example, 5760 × 1440 drops per inch), low-cost printing and production and good repeatability (range ~300 μm). Professional inkjet printers provide high spatial resolution and low production cost, are compatible properties with several materials and show repeatability ranges from 5 μm to 25 μm. Finally, mixed-screen printing and low-cost inkjet have demonstrated adaptabilty to numerous materials, good spatial resolution and repeatability (~300 μm) [57].

There are a few reported studies regarding the development of multilayered structures based on the development of ink properties such as viscosity, surface tension and pH combined with printing parameters (voltage and duration) [58]. Control over the evaporation rate is a key parameter for increasing accuracy and resolution, and the rate can be adjusted by the addition of cosolvents (e.g., alcohol)

### 1.2. Inks for Inkjet Printing

The ink used for material deposition and its physical properties is considered to be the most crucial part of inkjet printing technology [28]. Inkjet printing involves various processing steps such as droplet generation, motion, interaction with substrate, and drying that are influenced by the quality and properties of the ink for successful printing [11]. The resolution, uniformity and quality of the patterns significantly depend on the viscosity and surface tension of the ink [19,46]. These two parameters can determine the three main steps of inkjet printing process [46]. The speed and accuracy of droplet ejection will decrease if there is an increase in viscosity. At high viscosity and drop rate, the printing process will fail as the ink might not move towards the ink cartridge swiftly to refill it in-between the jetting [19]. Glycols (e.g., glycerol, propylene glycol and polyethylene glycol) are the typically used excipients for viscosity adjustment [59–61]. Contrastingly, surface tension influences the propensity of the ink to draw off of the nozzle to produce a droplet, and it is usually adjusted by adding surfactants [19]. Fromm et al. introduced an equation to determine an apt balance among the physical properties in one of his published articles in 1984, which is as follows:

$$Z = \frac{\sqrt{\gamma \rho l}}{\eta} = \frac{\sqrt{We}}{Re} = \frac{1}{Oh}$$

In this equation, the surface tension, density, viscosity and printing mesh aperture diameter are denoted by $\gamma$, $\rho$, $\eta$ and $l$, respectively [27,62]. This equation also explains the dimensionless numbers from fluid physics. i.e., Reynolds number, Weber number and Ohnesorge number designated by Re, We and Oh, respectively. It was suggested by Fromm that proper inkjet printing would be possible if $Z$ is greater than 4 ($Z > 4$). Further investigations led to the introduction of a limiting range for inkjet printing where $1 < Z < 10$ [27,63]. This range explains the viscous dissipation of a droplet formation if $Z$ is greater than 1 ($Z > 1$), and if $Z$ is less than 10 ($Z < 10$), then the formation of satellite droplets occurs (also termed as secondary droplets) [27]. These satellite droplets have an effect on primary droplets and influence their positioning on the substrate. In fact, the droplet deposition should be accurate, uniform and precise to facilitate successful inkjet printing [1]. Further

experimental studies revealed a new range for *Z*, that is 1 < *Z* < 14 [62,64,65]. Figure 4 shows the formation of satellite droplets.

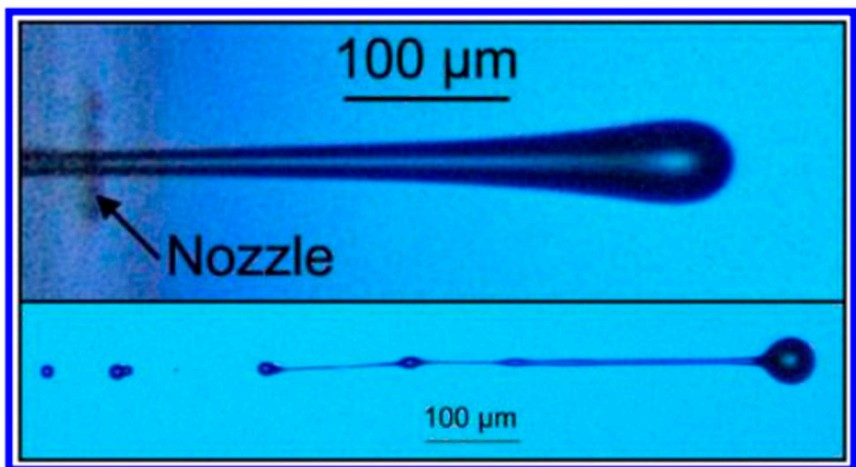

**Figure 4.** The upper image is a high-speed photograph of a droplet coming out of a nozzle, and the bottom one is a high-speed photograph of a satellite droplet formation [27].

The modulation of droplet size is a major challenge in inkjet printing. To date, a total of eight mechanisms have been recorded that are capable of changing the droplet volume utilizing same ink and printhead.

Usually, optical techniques are used to measure the droplet coming out of the nozzle [66]. To illustrate, a 6 ns short illumination with a laser induced fluorescent stroboscopic recording using iLIF (illumination by laser induced fluorescence) or ultra-high-speed cameras (up to 25 Mfps) are usually used to measure the drop formation with pL sized droplets at approx. 100 kHz repetition rate [67].

One of the inkjet printing limitations is the use of inks of low viscosities. Table 2 lists the compositions of some typically used printing inks including their viscosity and surface tension.

**Table 2.** Some of the recently used inks in inkjet printing.

| SN | Ink | Ink Viscosity (mPa·s) | Ink Surface Tension (m Nm$^{-1}$) | Z Value | Ref. |
|----|-----|----------------------|-----------------------------------|---------|------|
| 1. | Ethylene glycol | 15.8 | 45.5 | 2.08 | [68] |
| | Ethylene Glycol: Water (5/95) | 1.16 | 69.5 | 33.2 | |
| | Ethylene Glycol: Water (10/90) | 1.47 | 68.9 | 26.1 | |
| | Ethylene Glycol: Water (15/85) | 2.32 | 67.7 | 16.5 | |
| | Ethylene Glycol: Water (25/75) | 2.72 | 67.0 | 14.1 | |
| | Ethylene Glycol: Water (50/50) | 5.05 | 46.7 | | |
| | Ethylene Glycol: Water (50/50) | 4.39 | 60.3 | 8.40 | |
| | Ethylene Glycol: Water (75/25) | 7.81 | 52.7 | 4.47 | |
| | Ethylene Glycol: Water (85/15) | 10.5 | 50.2 | 3.28 | |

**Table 2.** *Cont.*

| SN | Ink | Ink Viscosity (mPa·s) | Ink Surface Tension (m Nm$^{-1}$) | Z Value | Ref. |
|----|-----|----------------------|-----------------------------------|---------|------|
| 2. | De-ionized water | 1.07 | 72.7 | 36.8 | [68] |
| 3. | Gallium-indium (75/25) | 1.7 | 624 | | [46] |
| 4. | Glycerol-Water | 1–22.5 | 66.4–7.6 | | [69] |
| 5. | CuNO4- Water | ~4.45 | 88 | | [70] |
| 6. | Dowanol | 10.17 | 15.55 | | [71] |
| 7. | Ethyl acetate | 0.452 | 2.367 | | [13] |
| 8. | 5 Fe$_3$O$_4$-95 (nanoparticles + UV Curable matrix resin) | 18.03 | 23.91 | 1.72 | [72] |
| 9. | 10 Fe$_3$O$_4$-90 (nanoparticles + UV Curable matrix resin) | 18.08 | 20.91 | 1.57 | [72] |
| 10. | Hydroxypropyl cellulose:Water (6/94) | 45 | 44.5 | | [73] |
| 11. | Commercial AgNp | 6.8 ± 0.7 | 30 ± 1 | | [74] |
| 12. | Diethylene glycol | 27.1 | 42.7 | 1.17 | [68] |
| 13. | Glycerol | 934.0 | 76.2 | 0.05 | [68] |
| 14. | MnCo$_2$O$_4$ | 10 | | 6.17 | [75] |
| 15. | MnCo$_{1.8}$Fe$_{0.2}$O$_4$ | >15 | | 4.77 | [75] |
| 16. | PVDF: BaTiO$_3$ (40/8) | 13.6 | 30.2 | 1.17 | [76] |
| | PVDF: BaTiO$_3$ (32/6.4) | 9.7 | 31.7 | 1.72 | [76] |
| | PVDF: BaTiO$_3$ (24/4.8) | 6.0 | 32.4 | 2.79 | [76] |
| | PVDF: BaTiO$_3$ (16/3.2) | 3.7 | 33.5 | 4.59 | [76] |
| | PVDF: BaTiO$_3$ (8/1.6) | 2.1 | 34.8 | 8.23 | [76] |
| | PVDF: BaTiO$_3$ (1/0.2) | 1.3 | 36.0 | 13.56 | [76] |
| 17. | DNTF: Hexogen (13.86/0) | 1.2 | 23.33 | 36.94 | [77] |
| | DNTF: Hexogen (12.47/1.39) | 1.0 | 23.09 | 44.56 | [77] |
| | DNTF: Hexogen (11.09/2.7) | 0.8 | 23.77 | 58.01 | [77] |
| | DNTF: Hexogen (9.70/4.16) | 0.6 | 24.15 | 75.51 | [77] |
| | DNTF: Hexogen (8.32/5.54) | 0.8 | 24.52 | 58.2 | [77] |
| | DNTF: Hexogen (6.93/6.93) | 1.3 | 23.66 | 35.44 | [77] |
| 18. | 8 mol% Y$_2$O$_3$-stabilized ZrO$_2$ (8YSZ) | 1.5 | 18.8 ± 0.3 | 7.6 | [78] |

Note: concentration of Ethylene Glycol: Water ratios are in *v*/*v*; concentration of PVDF: BaTiO$_3$ ratios are in mg mL$^{-1}$; concentration of DNTF: Hexogen ratios are in wt%; PVDF = Polyvinylidene difluoride; DNTF = 3,4-dinitrofurazanofuroxan.

### 1.3. Overview of Different Types of Inkjet Printing Technology

Based on the physical process of droplet generation, this automated, high-throughput technology is predominantly classified into two categories: (a) continuous inkjet printing (CIJ) and (b) drop-on-demand printing (DOD) (Figure 5) [16,23,27,29,79]. Continuous inkjet printers generate droplets as a continuous stream of ink discharged on the target, while in drop-on-demand printers, the droplets are ejected in a discontinuous manner only when they are needed [80]. Droplets deposited by continuous inkjet method are usually twice the size of the orifice [27].

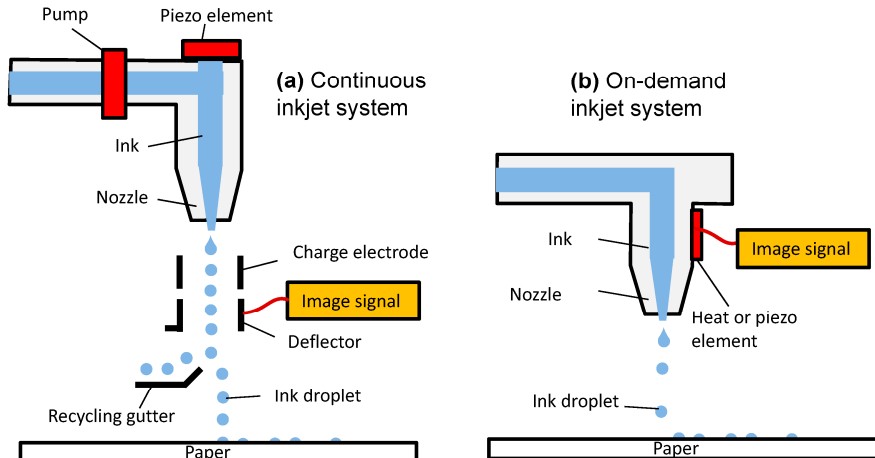

**Figure 5.** Simplified representation of two different categories of inkjet printing mechanism: (**a**) continuous inkjet printing (CIJ) and (**b**) drop-on-demand printing (DOD) [81].

In CIJ printing, a high-pressure pump under an electric field allows the continuous flow of liquid material that is ejected via the orifice, diameter 50–80 µm, which then disintegrate into a stream of droplets under the surface tension forces due to Reayleigh instability [13,15,79]. Continuous inkjet printing is predominantly used in textile printing, labelling and other high-speed graphical works [28]. Depending on actuation technique, IJP can be classified into another two categories which are: (a) piezoelectric and (b) thermal [82–85]. Both of the techniques reserve the material to be printed in a chamber and emit the droplets through the printhead via a nozzle, but they differ in the process of droplet formation [86–88].

In DoD inkjet printing, the liquid droplet is emitted through a nozzle only when it is required. Typically, a DOD printhead consists of multiple nozzles (usually 100–1000, aside from specialized printheads, which might have only one nozzle). The formation of droplets occurs swiftly after the deformation of the piezoelectric wall, which compresses the ink due to the applied wave. The ink material comes out of reservoir as a form of jet through the printhead, gravitates down afterwards and gets ejected via the nozzle under the surface tension forces to generate one or more droplets [15].

In contrast to CIJ, droplets produced by DOD inkjet printing are comparable with the diameter of the orifice, usually ranging from 10 to 50 µm, in accordance with drop volumes, which vary from 1 to 70 pL [15,27]. Due to the capability of smaller droplet formation, it has become a method of choice for several studies [28].

Biological ink materials can be affected by the electrostatic inkjet process due to the shear pressure (sonication with the frequency of 15–25 kHz), and they can clog easily since the diameter of the nozzle is not only fixed but also small [89].

The use of higher amount of solid material in the ink solution can increase the printing efficiency and decrease the cost notably which is one of the advantages of thermal inkjet printing [79].In addition, inks comprising aqueous solvents are usually more feasible for jetting with thermal inkjet printing. In contrast, organic solvents are generally more suitable for piezoelectric inkjet processing. Furthermore, thermal inkjet printers are usually inexpensive compared with the piezoelectric devices [1,90].

There are several other inkjet printing processes such as electrostatic, electrohydrodynamic and acoustic, but they are not frequently used because of their major drawbacks [82]. Some disadvantages of electrostatic inkjet printing are that it requires high voltage (sometimes over 2 kV) to operate, utilizes conductive metal pipe, requires placing one electrode externally to the device and requires placing a substrate between the nozzle and the electrode [91].

One of the drawbacks of electrohydrodynamic inkjet printing is that it cannot deposit single droplets at a time. The droplet generation occurs using an electric field and not by shrinkage of the ink with thermal energy or chamber deformation [89]. The low throughput

(low production speed) of electrohydrodynamic inkjet printing is considered to be the most severe drawback that has retarded its widespread application [92,93]. Figure 6 illustrates an overview of the available inkjet printing methods.

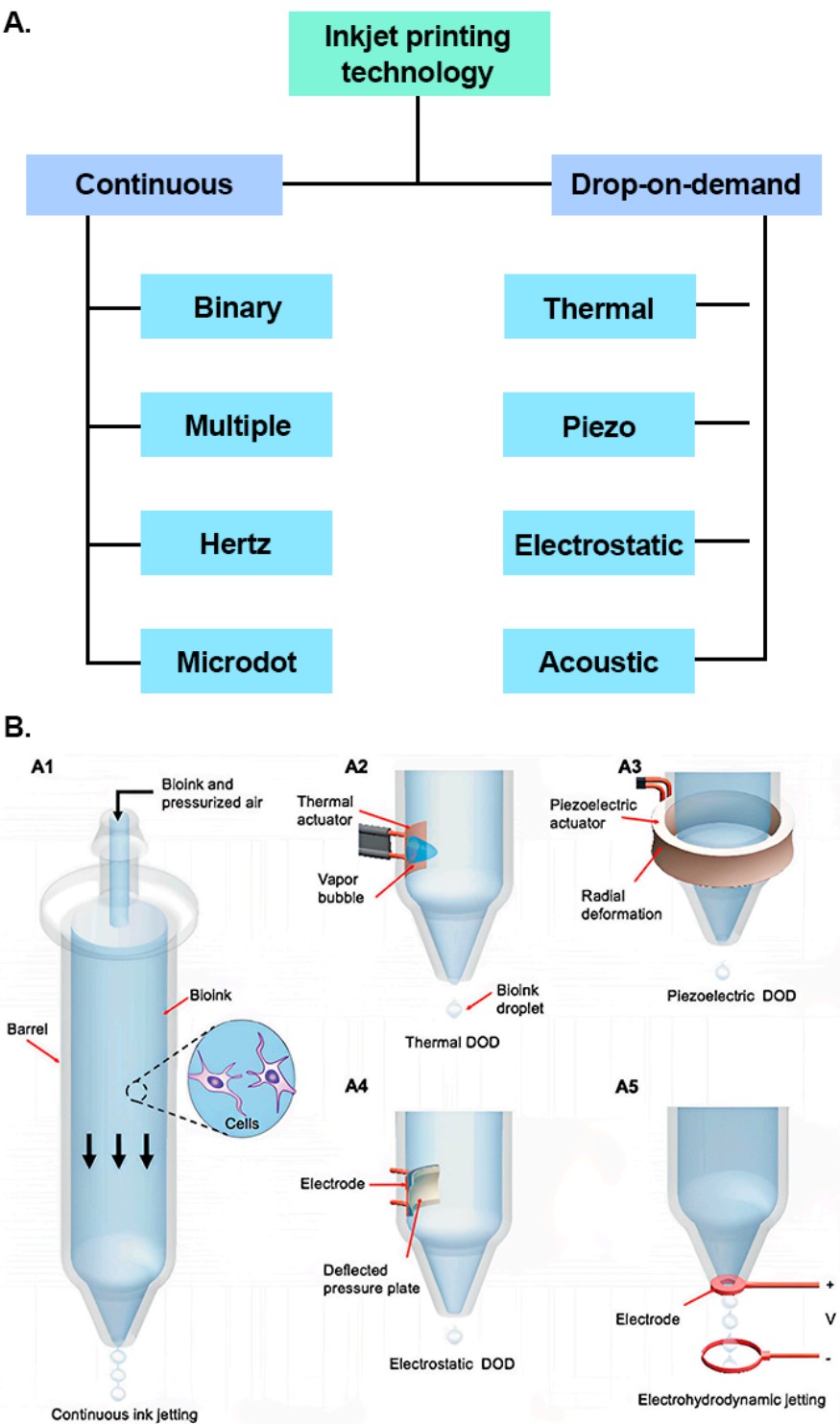

**Figure 6.** (**A**) Classification of IJP technology (reproduced from [94]). (**B**) Working principles of some of the different IJP techniques: (**A1**) continuous, (**A2**) thermal, (**A3**) piezoelectric, (**A4**) electrostatic, (**A5**) electrohydrodynamic (modified from [95]).

## 2. Thermal Inkjet Printing

Thermal inkjet printing (TIJP) is a noncontact DOD printing system that was basically developed for printing digital data on media [90,96,97]. It is also known as bubble inkjet printing since the droplet ejection occurs via bubble nucleation [28,87,98]. TIJ printers can eject droplets in a range of 2–180 pL of volume [99,100].

A TIJ printer consists of an ink (desired fluid material to be printed), the cartridge and a printhead. The printhead comprises several nozzles (column like small channels) filled with the fluid material from the ink chamber, and a transducer (which is a thin film resistor for thermal inkjet printing) is attached to each nozzle [79,100].

Due to its reproducibility, low cost and high throughput, this printing technique dominates the market over other printing technologies [47,48,101–104].

In TIJP, a thin-film resistive heater that creates a frequency ranging from 1 kHz to 5 kHz with an approximate rectangular wave of 3–6 μs pulse width is attached to the printhead, which instantaneously heats the ink in the cartridge. A small vapor bubble forms and puffs up using the heat, which generates a pressure pulse essential for droplet emission through the nozzle. Once a droplet emission is completed, the current is withdrawn, which facilitates a prompt reduction in the vapor pressure and temperature. Consequently, the bubble collapses inside the printhead, which somewhat creates a vacuum (negative pressure) that pulls the liquid ink to refill the chamber [4,47,80,98,105–111]. Thermal gradient, viscosity of ink material and electric pulse frequency determine the size of the droplets to be generated [22,47,109,111].

In TIJP, the thermal resistor can momentarily (approximately 3 to 10 μs) produce up to 300 °C temperature, and merely around 0.5% of the ink encounters a thermal rise in the nucleation of a vapor bubble [89,105].

*Types of Thermal Inkjet Printer*

Depending on the droplet emission principle, there are three types of TIJ printer available: (a) side shooter, (b) roof shooter and (c) back shooter [13,48]. For the side shooter printer, the droplet is ejected tangential to the surface of heater. On the other hand, the droplet ejects straight (at 90° angle to the heater surface) in a roof shooter. In the back shooter printer, the droplet ejects straight, but the vapor bubble nucleation occurs in the opposite direction of droplet emission [48]. Simplified versions of the working mechanisms behind the side shooter, roof shooter and back shooter are shown in Figure 7.

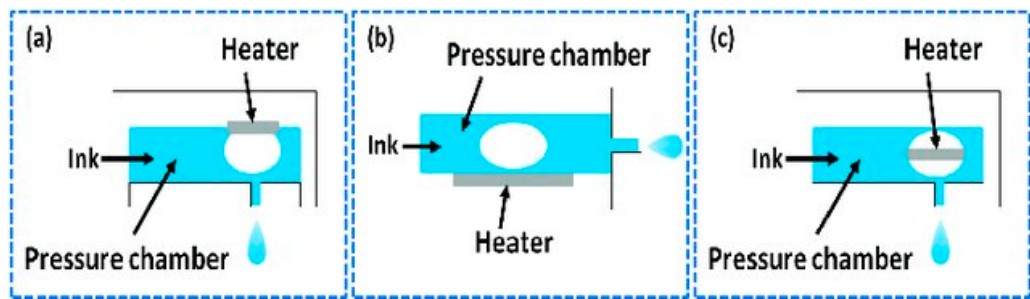

**Figure 7.** (**a**) Roof shooter, (**b**) side shooter, and (**c**) back shooter system [13].

## 3. Application of Thermal Inkjet Printing

TIJP has found several applications for printing of medicines in various forms by controlling the printing pattern and the deposited amount, most importantly for the printing of a wide range of drug formulations. In addition, it has found a strong ground in bioprinting applications for cell-laden bioinks in tissue engineering and regenerative medicine.

### 3.1. Bioprinting

An engineered tissue can be used as a physiological replica for better understanding of basic biology. Moreover, the problem with finding suitable organ donors for repairing or replacing damaged or injured organs, regenerative medicine, cell transplantation and tissue engineering may be resolved by using bioprinting technology (see Figure 8) [84,112–115]. Thermal inkjet printing-based bioprinting is one of the promising approaches in the field of tissue engineering [111]. There are some other bioprinting strategies apart from TIJP, for instance, extrusion bioprinting (mechanistically similar to traditional fused deposition modeling (FDM) 3D printing, which extrudes droplets via pneumatic pressure or mechanical forces through a nozzle onto a previously fixed location called the fabrication platform) [116] and vat polymerization-based bioprinting (mostly used for tissue scaffold fabrication utilizing conventional cell-seeding approach) [117].

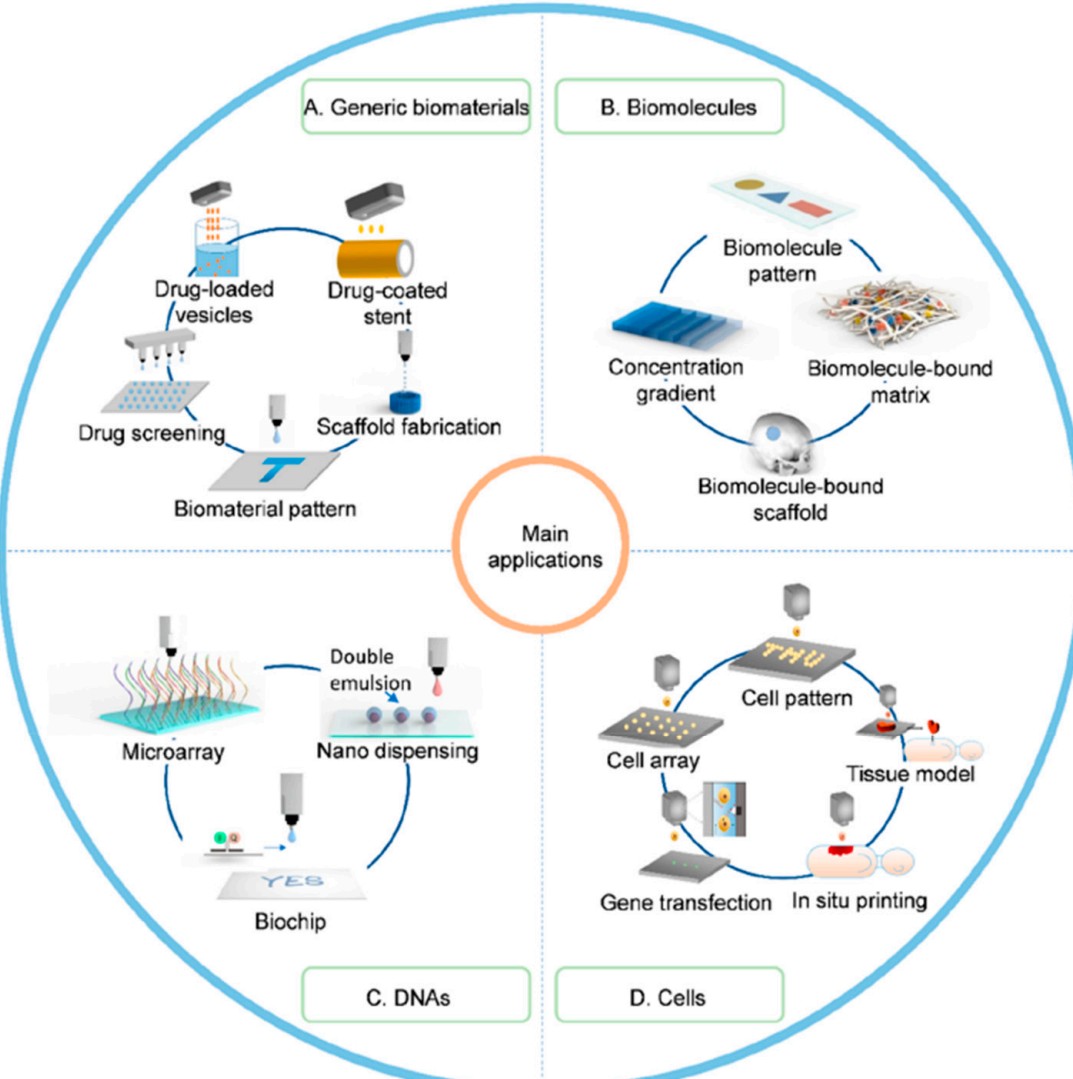

**Figure 8.** Schematic illustration of major applications of inkjet bioprinting. (**A**) Generic biomaterials were printed for pharmaceutical applications including drug screening, drug loading and drug coating and for biomaterial patterning and scaffold fabrication. (**B**) Biomolecules were printed for concentration gradient, biomolecule pattern, biomolecule-bound matrix and biomolecule-bound scaffold. (**C**) DNAs were printed for microarray, nano-dispensing and biochip. (**D**) Cells were printed for cell array, cell pattern, tissue model, in situ printing and gene transfection [89].

Printers used in bioprinting fabricate the biological elements in a layer-by-layer manner often described as a bottom-up process since the deposition of the first layer is followed by the buildup of more layers. [82,106]. In bioprinting, the conventional ink material is replaced with bioink, which is a liquid and contains water [112] along with biological substances, e.g., enzymes [118], proteins [65], saline or other media with suspended cells [119]. Some studies observed a 10–15% decrease in the enzyme activity [111,120], and hence, an increase in temperature for bubble nucleation was suspected to affect the biological substances in the bioink chamber [47,106,111]. To overcome this issue, heat is transmitted for merely 2 μs, which causes a 4–10 °C increase in temperature that ensures a count of 90% viable cell, enough to conduct bioprinting efficiently [89,106,111,112].

Despite the significant number of studies, there is still a lack of understanding of how the viability of human primary cells is affected during the inkjet printing of sub-nanolitre amounts. In a recent study, Ng et al. used TIJP to dispense cell-laden inks to investigate cell viability and proliferation [121]. It was observed that increased cell concentrations had a minimal impact on the droplet velocities but led to better cell viability. By regulating the droplet volumes at 20 nL, it was possible to eliminate the evaporation-induced damage to the cells, which also resulted in high viability.

Park et al. developed a strategy for the formation of self-organized 3D collagen microstructures [122]. By applying DoD inkjet printing with predefined patterns, it was revealed that cell-to-extracellular matrix interactions facilitate the self-organization of microstructures on hydrogels comprising collagen, while actin polymerization inhibits the formation of the microstructures. Further manipulation of the print patterns and cell densities assisted with the formation of a human skin model with papillary microstructures.

Suntornnond et al. introduced significant advances in TIJP by expanding the use of printable bioinks [123]. The authors applied saponification in gelatin methacrylate (GelMA), a very common bioink, to study the printability in a thermal inkjet printer (HP Inc. D300e Digital Dispenser). The two-step process, which comprised saponification and heat treatment, led to the formation of excellent bioinks with good cell viability and proliferation. Saponification is an exothermic reaction where the hydrolysis of triglycerides with alkali produces salts of fatty acid and glycerol [124,125]. Sun et al. presented a detailed description of the saponification process that was investigated via isothermal titration calorimetry, attenuated total reflection infrared and small-angle X-ray scattering spectroscopies [124]. Another group of researchers, Tan et al., published a review in 2012 on glycerol (which is a valuable byproduct of the saponification process) in which they discussed different methods of glycerol production as a byproduct and its purification process [125]. In bioprinting, both saponification and heat treatment are usually used to enable better jetting behavior of the ink.

Yoon et al. employed the saponification of gelatine methacryloyl to stabilize the jet formation and to reduce the viscoelasticity of the inks [126]. The use of inkjet printing allowed for the large-scale production of multilayers and ensured high shape fidelity. The use of alginate, cellulose nanofiber, fibrinogen blended with human dermal fibroblasts facilitated the generation of structures that mimic native tissue functions. Freeman et al. (Figure 9) mentioned that heat treatment of their material (gelatin) induced favorable rheological properties that increased the printability of ink; however, it also affected the cell viability, which they optimized by increasing cell density and tissue volume, and that ultimately increased the collagen deposition and mechanical strength of printed material [127].

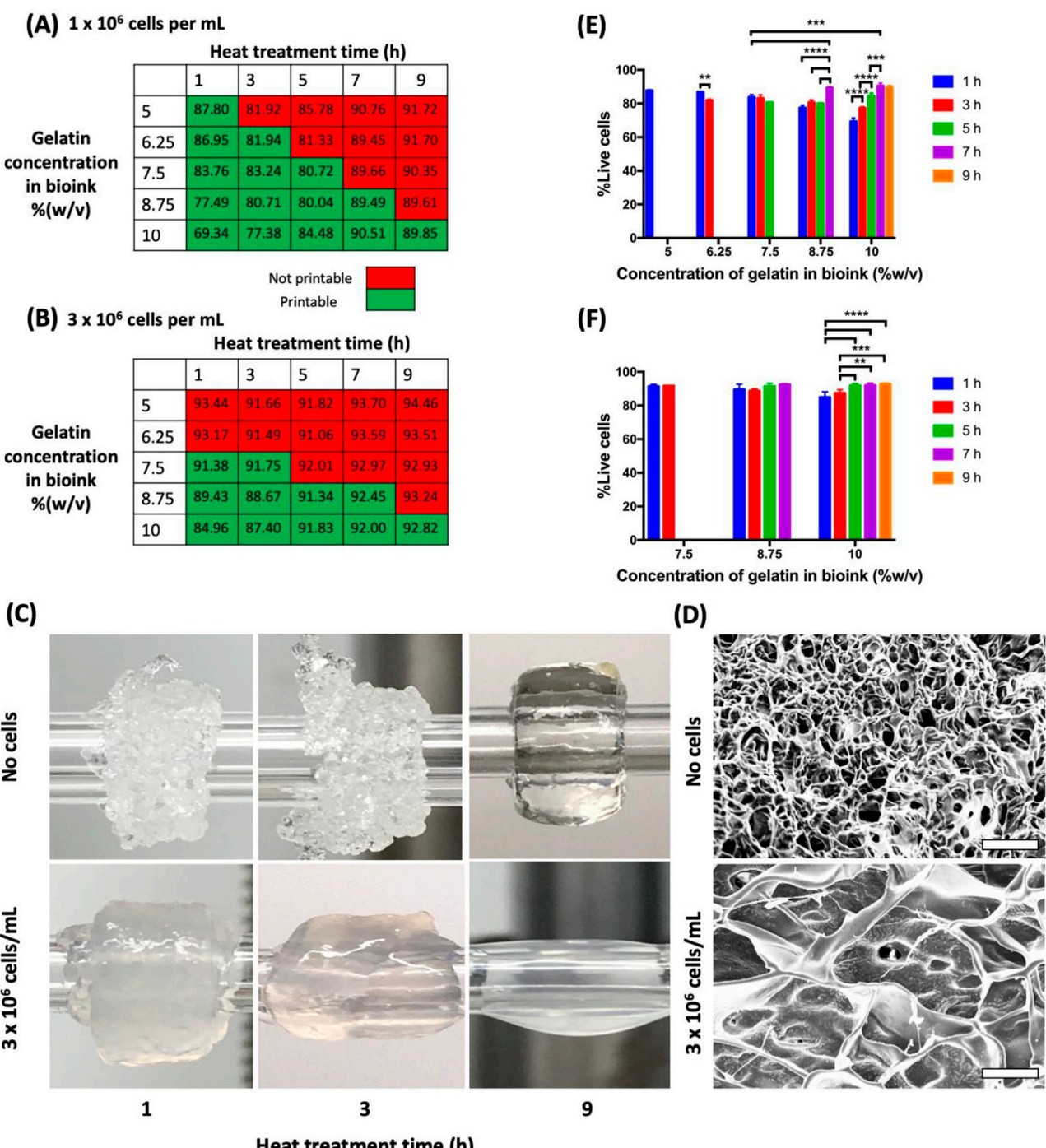

**Figure 9.** Effect of heat treatment of gelatin on cell viability during 3D rotary printing. (**A**) $1 \times 10^6$ or (**B**) $3 \times 10^6$ cells/mL neonatal human dermal fibroblasts (HDF-n) was mixed with the gelatin–fibrinogen bioink for vascular 3D rotary printing. The gelatin was heat treated at 90 C before use in preparing the bioinks. The red regions indicated poor printability of the cell-laden bioinks, while the green regions indicated conditions that held. Average percentage of live cells detected by ethidium homodimer staining are shown in the table. (**C**) The appearance of tubular tissue constructs printed using cell-laden bioinks prepared using 10 mg/mL of fibrinogen and 7.5% ($w/v$) heat-treated gelatin. (**D**) SEM micrographs of the 5% ($w/v$) 1 h heat-treated gelatin + 10 mg/mL fibrinogen. Scale bar: 10 mm. Percentage of live cells in gelatin–fibrinogen bioinks containing (**E**) $1 \times 10^6$ cells/mL or (**F**) $3 \times 10^6$ cells/mL. Data are presented as mean ± S.D. **, $p < 0.0021$; ***, $p < 0.0002$; and ****, $p < 0.0001$ [127].

Solis et al. employed TIJP to develop human microvascular endothelial cells for the formation of microvasculature to print implantable graft–host anastomoses [128]. Flow cytometry revealed 75% apoptosis during printing, but the viability improved after 3D incubation. Further investigations showed the overexpression of various cytokines such as HSP70, IL-1α, VEGF-A, IL-8 and FGF-1 for the printed cells. The activation of the HSP-NF-κB pathway led to the production of VEGF and consequently to the immense formation of capillary blood vessels after implantation.

Figure 10 shows another example of how TIJP was used for tissue engineering and regeneration by Gao et al. [129]. The authors used a TIJP with continuous photopolymerization to develop a bioprinting platform for 3D cartilage tissue engineering. The created cartilage showed native zonal arrangement with excellent extracellular matrix architecture, and the required material performance. The vitality of the printed cells with concurrent photopolymerization was noticeably higher than with the control tissue creation method, which necessitates prolonged UV exposure. Substantial glycosaminoglycan (GAG) and collagen type II production was seen in printed neocartilage that was compatible with the gene expression pattern.

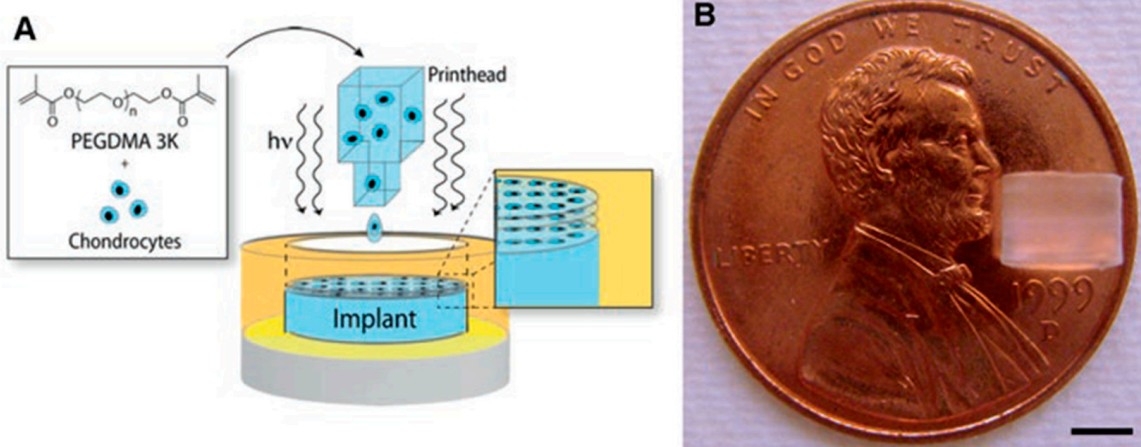

**Figure 10.** Bioprinted neocartilage tissue using TIJP with continuous photopolymerization. (**A**) Schematic of cartilage bioprinting with simultaneous photopolymerization and layer-by-layer assembly. (**B**) A printed neocartilage tissue of 4 mm in diameter and 4 mm in height. Scale bar, 2 mm [130].

In another study, Kador et al. combined the formed electrospun cell transplantation scaffolds with retinal ganglion cells and positioned them accurately on the scaffolds using thermal inkjet printing. This procedure preserved the printed cells' functioning electrophysiological capabilities, cell growth, and neurite expansion [131].

Furthermore, Gao et al. used TIJP to determine the effectiveness of bioactive ceramic nanoparticles in promoting osteogenesis in human bone marrow mesenchymal stem cells that were printed on poly(ethylene glycol) dimethacrylate scaffolds [132]. The printing of the stem cells suspended in poly(ethylene glycol) dimethacrylate scaffolds with nanoparticles of bioactive glass and hydroxyapatite during simultaneous polymerization enabled the deposition of the printed substrates with extremely precise placement in 3D locations. Further analysis revealed that the printed scaffolds produced far more collagen and had the most alkaline phosphatase activity, comparable with the gene expression found by quantitative PCR. The study was an example of how inkjet printing can be employed for the engineering of both soft and hard tissues with biomimetic assemblies.

As summary of various studies that have used thermal inkjet printing is presented in Table 3, illustrating the different applications and the main positive or negative outcomes reported for each work.

**Table 3.** List of thermal inkjet-based bioprinters with their applications & outcomes in biopharmaceutical research area.

| SN | Printer | Bioink | Area of Application | Outcome (Positive, Negative or Both) |
|---|---|---|---|---|
| 1. | HP Deskjet 500 printer (modified) [Hewlett-Packard, Inc., Palo Alto, CA] | Rat tail collagen type I | Cell printing | Around 89% cell viability was reported [111]. |
| 2. | HP DeskJet 550C printer (modified) | hAFSCs cell line | Stem cell printing | Data revealed that printed hAFSCs are capable of forming a firm bony tissue that can withstand high compressive force [107]. |
| 3. | Prototype of thermal inkjet printer combined with amperometric GOD electrode [developed by Lesepidado srl (Bologna, Italy) & supplied by Olivetti Tecnost (Ivrea, Italy)] | Glucose oxidase (GOD) from *Aspergillus niger* and poly(3,4-ethylene di-oxy thiophene/ polystyrene sulfonic acid) | Biosensor | Approximately 15% decrease in the efficiency of enzyme was noted [120]. |
| 4. | Canon inkjet printer (Pixma ip4500) (modified) | Fluoroscein isothiocyanate-conjugated bovine albumin and horseradish peroxidase | Microfluidic patterned paper | Bioactivity was retained by patterned paper. However, the percentage was not measured [118]. |
| 5. | Hewlett-Packard (HP) Deskjet 560 (Modified) | Herring sperm DNA in pure water, surfactant, alcohol, or a water-soluble polymer | Microarray | Was reported as a dependable printing option [133] |
| 6. | Bubble Jet (BJC-2100, Canon, Tokyo, Japan) | Rat tail collagen solution | Cell patterning | Spatial resolution of around 350 mm was obtained, and adherence of neuronal and smooth muscle cells to the printed area was reported [134]. |
| 7. | BJ F850 (Canon, Tokyo Japan) | Insulin related growth factors | Cell patterning and analysis | Intensified proliferation of cells on patterned area was observed [133]. |
| 8. | HP Deskjet 500 inkjet printer (modified) [Hewlett-Packard, Inc., Palo Alto, CA] | Chinese hamster ovary (CHO) cells | Cell patterning | Cellular viability count of 80% was reported that improved after changing the carrier fluid. Transient membrane damage of cells was observed after printing [111]. |

**Table 3.** *Cont.*

| SN | Printer | Bioink | Area of Application | Outcome (Positive, Negative or Both) |
|---|---|---|---|---|
| 9. | HP DeskJet 692C and 55uC | CHO cells and porcine aortic endothelial | Gene transfection | Transfection rate of 10% and cellular viability of 90% were reported [135]. |
| 10. | HP Desktop printer (HP 55uC) (modified) | Mouse myoblast | Biosensor | Myotube generation alongside printed substrate was demonstrated [136]. |
| 11. | Hewlett Packard (HP) Deskjet 500 | Mammalian cells | Cell printing | Cellular viability varied 85–95% [112]. |
| 12. | HP-2225C Think Jet ink jet printer 7470A graphics plotter | Fibronectin | Cell patterning | Stickiness of cells to patterned fibronectin was noticed [137]. |
| 13. | BJC-600 (Canon, Tokyo) and BJC-700J printer | 5′-terminal-thiolated oligonucleotides | DNA microarrays | No trouble was encountered by researchers while ejecting DNA solution using bubble jet printer rather than heat generation, which was stated as an added advantage as it provided efficient reaction energy [138]. |
| 14. | Prototype model of TIJ printer from Olivetti Tecnost developed by Lesepidado srl | β-Galactosidase (GAL) from *Aspergillus oryzae* | Biosensor | Aside from approximately 15% reduction in enzyme activity, TIJP was determined to be a promising option for enzyme or other biological material micro-deposition [110]. |
| 15. | HP60 inkjet printer | Unmentioned cell | Cell printing | Successful concurrent simulation of thermal transfer, interaction between cell and fluid, transition of phase and increased cell viability was reported [47]. |
| 16. | Hewlett Packard Deskjet 500 thermal inkjet printer (modified) [Hewlett–Packard Company (Palo Alto, CA, USA)] | Human microvascular endothelial cells (HMVEC) and fibrin | Cell printing | Printed HMVEC proliferated and the formation of microvascular endothelial cells along with fibrin scaffolding was observed [139]. |

**Table 3.** *Cont.*

| SN | Printer | Bioink | Area of Application | Outcome (Positive, Negative or Both) |
|---|---|---|---|---|
| 17. | Canon inkjet printer (Pixma ip4500) (modified) | Horseradish peroxidase (HRP) and alkaline phosphatase (ALP, from bovine intestinal mucosa) | Enzymatic paper | Bioactivity was retained by patterned enzymatic paper, but the percentage was not mentioned [140]. |
| 18. | Hewlett Packard (HP) 550C printer (modified) | Suspensions were made using embryonic motoneurons of rat and Chinese Hamster Ovary (CHO) | Cell printing | Successful printing of embryonic motoneurons and CHO cells with >90% viability was reported [104]. |
| 19. | Hewlett-Packard (HP) Deskjet thermal inkjet printer | Bone-marrow derived hMSCs | Cell printing | Viability of the printed cells was significantly higher [129]. |
| 20. | HP TIPS print head (Hewlett-Packard Packard, Corvallis | Retinal ganglion cells | Cell printing | Comparatively better cell survival, neurite outgrowth and functional electrophysiological properties of the printed cells were observed [131]. |

*3.2. Oral Dosage Form*

Today, most of the market oral drug products are in the form of tablets or capsules where more than 40% of the APIs (active pharmaceutical ingredients) are water insoluble [41]. The norm for the increase of drug solubility and hence the bioavailability [103,104] involves a range of processing technologies such as particle size reduction, salt formation, cocrystals, granulation, or solid dispersions [141,142].

As previously mentioned, those technologies are designed for mass production and are not suitable for the design and administration of personalized dosage forms that address the specific needs of individual patients such as children or elders. Over the last 10 years, thermal inkjet printing has been adopted as an ideal technology (Figure 11) for the printing of unique dosage forms with various features. [105,143]. A well-known technology named binder jetting (a nonfusion powder additive manufacturing technology) has been implemented for the printing of pharmaceutical drug dosage form [144]. Aside from the FDA approval for the Spritam (a binder jet 3D printed dosage form) [145], there have been experimental approaches using binder jetting technology. For example, Chang et al. used binder jet 3D printing (BJ-3DP) to print solid dosage forms. They used feedstock materials of pharmaceutical grade to make printed tablets and also developed a molding method for the selection of suitable powder and binder materials [146]. Rahman et al. discussed that the advantages of using binder jet 3D printing are that no polymers with special properties are needed, and any available FDA-approved excipients can be used for dosage form preparation [147]. Hong et al. used BJ-3DP to develop multicompartmental structure-dispersible tablets of levetiracetam-pyridoxine hydrochloride (LEV-PN), and they also managed to trounce the coffee ring effect by modifying the drying method in their study [148]. Kozakiewicz-Latała et al. developed a fast-dissolving tablet using BJ-3DP. Here,

they used two easily available FDA-approved model APIs, (a) quinapril hydrochloride (hydrophilic, logP = 1.4) and (b) clotrimazole (hydrophobic, logP = 5.4) [149].

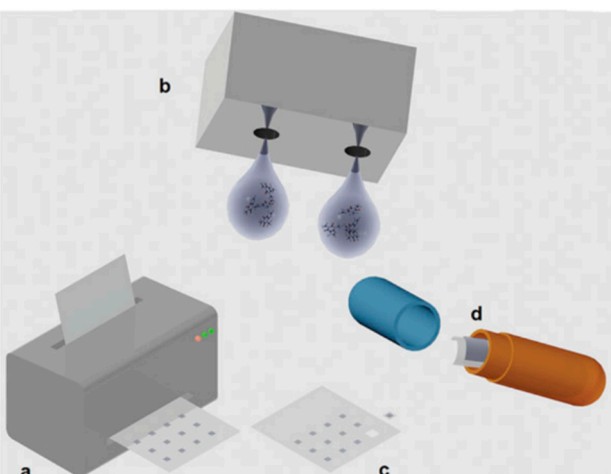

**Figure 11.** Schematic demonstration of the printing concept for pharmaceutical oral dosage form: (**a**) inkjet printer, (**b**) therapeutic material deposition, (**c**) unit doses on a paper substrate and (**d**) doses inserted to be into capsules or directly fabricated into oral dosage forms [61].

TIJP has demonstrated excellent capability of producing drug crystals rapidly and generating fine particles by dispensing volumes of drug solutions ranging from 5 to 15 pL. A Hewlett-Packard HP460 Deskjet thermal inkjet printer was used for the formulation of orally administrable co-crystal dosage forms. Various solutions of carbamazepine, nicotinamide, benzoic acid, isonicotinamide, theophylline and saccharin were printed and evaluated for their capacity to produce co-crystals using water/ethanol inks [150].

In another study, TIJP was employed for the printing of APIs on a substrate followed by polymer coating. As ink material, riboflavin sodium phosphate or paracetamol were dissolved in a glycerol and purified water solution following printing at two different dose intensities [151]. Wilts et al. used a combination of a thermal inkjet printhead HP$_{11}$ and ZCorp Spectrum Z510 printer to formulate acetaminophen tablets through the comparison of 4-arm star and linear poly(vinyl pyrrolidone) as binder materials [152]. The molecular weight and polymer concentration were found to affect the ink jetability, tablet porosity, hardness and drug loading on the tablet.

TIJP has been successfully used for the printing of prednisolone, a poorly water-soluble drug, in polymorphic forms. Using a mixture of glycerol, water and ethanol at a ratio of 3:17:80, the drug droplets were deposited on substrates comprising fiberglass films [86]. Raman analysis showed that the selection of the ink solvents was the main reason for the formation of the prednisolone polymorphs on the substrate, and this could be important for the design of oral solid forms and the printing of the most stable polymorphs with the fastest dissolution rates.

Montenegro-Nicolini and coworkers introduced inkjet printing for the deposition of biological molecules and the formation of lysozyme [153]. Polymeric films of hydroxypropyl methylcellulose and chitosan were further developed by applying polycaprolactone fibers using electrospinning prior to the molecule deposition. The printed drug amounts did not affect the muco-adhesiveness or mechanical properties of the films. The same group used a Hewlett Packard Deskjet 1000 to print buccal dosage forms comprising lysozyme and ribonuclease-A [154]. Printing proteins and peptides is challenging, but the use of thermal inkjet printing was proved a promising approach.

Buanz and coworkers used a modified cartridge of a HP Deskjet D1660 TIJ printer to print oral thin films of salbutamol sulphate. Potato starch was used as substrate for the dispensing of salbutamol sulphate dissolved in distilled water and glycerin in a series

of different concentration ratios [100]. The surface tension had no effect on the droplet deposition, while the calibration of the printing process allowed the design of personalized dosage forms. The same group used a modified Hewlett-Packard HP 5940 Deskjet to formulate orodispersible vitamin films (ODF) of warfarin at dose strengths varying from 1.25–2.5 mg [99]. The film substrates comprised HPMC/glycerol blends and showed rapid disintegration. The study demonstrated the capabilities of inkjet printing for the development of personalized dosage forms for API with narrow therapeutic index. Wickström et al. used an unmodified TIJ Canon Pixma desktop printer to print ODFs for pediatric administration [33]. A multicomponent formulation comprising B, B1, B2, B3, and B6 vitamins was printing on rice paper (Easybake® edible rice paper) that was used as a film substrate. The technology was validated for the accuracy and reproducibility of the printed doses.

More recently, Tam et al. developed paracetamol ODFs for point-of-care applications using HPMC as ink component [155]. The technology used for this study differed from typical thermal inkjet printing as it incorporates a piezoelectric micro-dispensing system to overcome viscosity limitations. Another advantage of the printed ODFs is that there is no need to use film substrates as they are formed during the dispensing. Even at high ink viscosities (32–818 mPa·s), it was possible to develop transparent films with homogenous distribution of materials.

Inkjet printing has been used for water-based inks and the development of pharmaceutical dosage forms [156]. A Fujifilm Dimatix printer was employed to develop polyvinylpyrrolidone and thiamine hydrochloride inks for the printing of tablet forms. Interestingly, the printing process optimization prevented the recrystallization of theophylline in the PVP matrix, and the dissolution rates were fast and were not related to the number of printed layers (Figure 12).

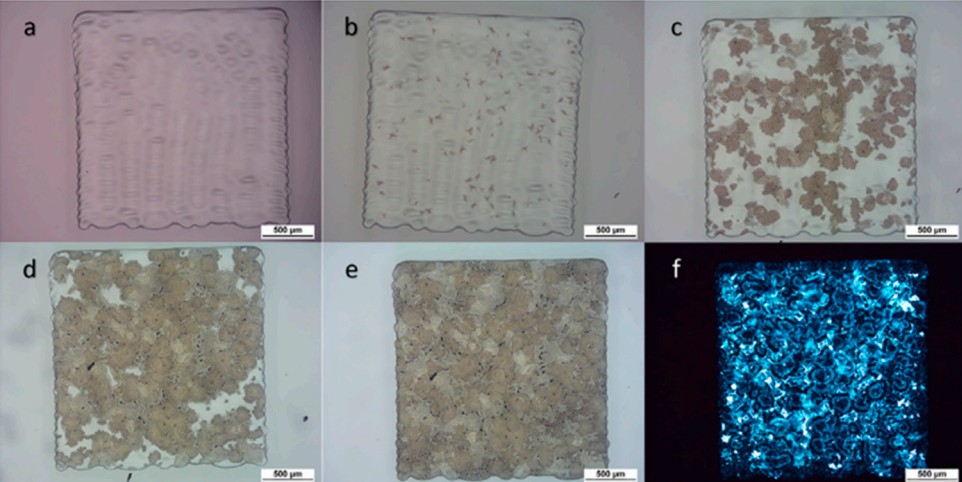

**Figure 12.** Optical reflection images of printed films on glass slide: (**a**) 30 min, (**b**) 1 day, (**c**) 4 days, (**d**) 8 days, (**e**) 12 days, (**f**) 12 days after orienting. All confirm the formation of a crystalline phase (cross-polarized transmission OM). Scale bar = 500 µm [156].

By developing macrogol inks on edible substrates, Thabet at al. prepared ODFs of hydrochlorothiazide and enalapril maleate [157]. The dynamic viscosities varied from 7 to 17 mPa.s through the addition of small ethanol amounts. More importantly, the careful selection of formulation composition prevented the drug's recrystallization on the edible substrates without affecting the mechanical properties as well.

Aerosol drug delivery has also been explored by scientists using TIJP technology. A combination of thermal inkjet printing and spray freeze drying (TIZ-SFD) was applied to formulate inhalable powders of terbutaline sulphate and compared with the marketed Bricanyl product [158]. By modifying a Hewlett-Packard thermal printer, it was feasible to atomize terbutaline sulphate (excipient-free) solution incorporated in liquid nitrogen

followed by freeze drying of the produced droplets afterwards. The process resulted in spherical and porous particles with a volume median diameter of 14.1 ± 0.8 µm and mass median aerodynamic diameter of 4.0 µm, respectively. The measured fines in the TIZ-SFD process were found to be 22.9%, in contrast with the 25.7 µm of the Bricanyl Turbohaler.

Similarly, it was demonstrated that TIZ-SFD is capable of processing up to 15% $w/v$ salbutamol sulphate (SS) solution and producing droplets of around 35 µm diameter. The samples analyzed with a twin-stage impinger showed 24.0 ± 1.2% and 26.4 ± 2.2% fine particle fraction. The result is scalable, and TIZ-SFD showed better outcome for inhalable particle formulation in comparison to standard Cyclocap® [159].

One of the great advantages of TIJP is the feasibility of processing a range of materials and hence producing multimaterial objects with complex geometries. Lion et al. investigated the development of multilayer dosage forms with tailored drug doses and release profiles using solvent-free thermal inkjet printing [160]. Using Compritol HD5 ATO as the core material, they printed multilayer structures of complex geometry (Figure 13) that could tune the release of Fenofibrate (loadings varied from 5–30%) and provide both sustained or immediate release rates. The printer features allow the production of droplets of 30 pL in volume. The printing consistency facilitated the fabrication of honeycomb internal integrity and various channel diameters.

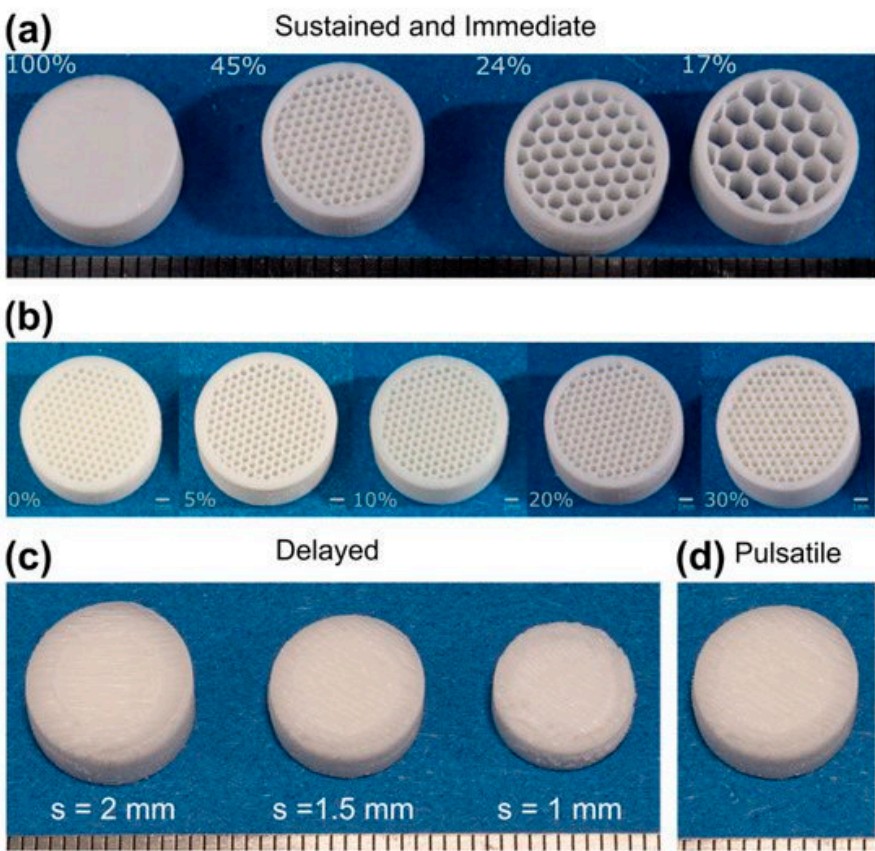

**Figure 13.** Images of sustained- and immediate-release tablets (**a**) with constant drug content (10% drug loading) and varying infill and (**b**) with varying drug content (5–30% drug loading) and constant infill (45%). (**c**) Delayed-release tablet with 1, 1.5 and 2 mm shell thickness (10% drug loaded core), and (**d**) pulsatile-release tablets. Ruler unit: 1 mm [160].

Table 4 summarizes typical examples of TIJP that have been investigated for printing oral solid dosage forms using thermal inkjet printing.

**Table 4.** TIJ printing examples used in printing of oral solid dosage forms.

| Sl No. | Printer | Dosage Form | Ink Material | Ref. |
|---|---|---|---|---|
| 1. | Hewlett-Packard HP460 Deskjet | Cocrystal | carbamazepine, nicotinamide, benzoic acid, isonicotinamide, theophylline and saccharin | [150] |
| 2. | HP Photosmart B010 | Cocrystal | riboflavin sodium phosphate and paracetamol | [151] |
| 3. | Combination of thermal inkjet printhead HP$_{11}$ and ZCorp Spectrum Z510 | Tablets | acetaminophen | [152] |
| 4. | Hewlett-Packard 970 Cxi DeskJet | Tablets | prednisolone | [86] |
| 5. | Hewlett Packard Deskjet 1000 | Buccal film | lysozyme | [153] |
| 6. | Hewlett Packard Deskjet 1000 | Buccal film | lysozyme and ribonuclease-A | [154] |
| 7. | HP Deskjet D1660 | Oral film | salbutamol sulphate | [100] |
| 8. | HP 5940 Deskjet | Orodispersible films | warfarin | [99] |
| 9. | TIJ Canon Pixma (unmodified) | Orodispersible films | vitamin B B1, B2, B3, and B6 | [33] |
| 10. | Nanojet Piezo Valve NJ-K-4020 | Orodispersible films | paracetamol | [155] |
| 11. | Fujifilm Dimatix DMP-2850 Series | Tablet | polyvinylpyrrolidone and thiamine hydrochloride | [156] |
| 12. | PIXDRO JS 20 | Orodispersible films | hydrochlorothiazide and enalapril maleate | [157] |
| 13. | TIZ-SFD | Powder particle | terbutaline sulphate | [158] |
| 14. | TIZ-SFD | Powder particle | salbutamol sulphate | [159] |

### 3.3. Antimicrobial Resistance Control

Antimicrobial therapies are not limited to bacterial infections [161,162] but are also used in the treatment of cancer [163–165], Alzheimer's disease [166] and other neurological disorders [167,168]. In this situation of the rapidly growing use of antimicrobial therapies, antimicrobial resistance is also increasing at a similar pace. Sadly, the dramatically increasing rate of antimicrobial resistance has become a global concern [169–172].

Therefore, to overcome the issues associated with the inappropriate administration of antimicrobial drugs, MIC (minimum inhibitory concentration) is being assessed these days [173–176]. MIC assessment as a quantitative analysis determines the lowest concentration of an antimicrobial therapeutic by which a specific microorganism's growth can be inhibited [175,177]. However, the conventional techniques available for MIC assessment such as agar and broth dilution and broth microdilution have drawbacks, i.e., trouble attaining different therapeutic concentrations in extensive scale and room for potential errors [97,176]. Hence, automated technologies like thermal inkjet printing are being explored in this area. Since any technique for MIC assessment should have the properties of accuracy, controlled deposition and high throughput, TIJP is the best match for performing MIC. Moreover, TIJP requires only tiny volume of sample which is complimentary in this case [101,133,178].

A group of researchers used a Hewlett Packard (HP) 5940 Deskjet thermal inkjet printer and broth microdilution for their studies to evaluate the MICs of a few antibiotics, amoxicillin, ampicillin, doxycycline and tetracycline (all of them were 92.5–100.5% pure) against Lactobacillus acidophilus. Data obtained from their experiment (see Table 5) show that the MICs for the tested antibiotics were within the acceptable range for TIJP, in contrast with broth microdilution [97].

**Table 5.** Calculated MICs of antibiotics against Lactobacillus acidophilus determined via thermal inkjet printing and broth microdilution.

| SN | Antibiotic | Thermal Inkjet Printed MIC(µg/ mL) | Broth Microdilution MIC(µg/mL) |
|----|-----------|-----------------------------------|-------------------------------|
| 1. | Amoxicillin | 0.20 | 0.5 |
| | | 0.23 | 0.5 |
| | | 0.15 | 0.5 |
| | | 0.19 | 0.5 |
| 2. | Ampicillin | 0.12 | 0.25 |
| | | 0.12 | 0.25 |
| | | 0.15 | 0.25 |
| 3. | Doxycycline | 0.29 | 1 |
| | | 0.31 | 1 |
| | | 0.29 | 1 |
| | | 0.35 | 1 |
| 4. | Tetracycline | 0.59 | 2 |
| | | 0.55 | 2 |

## 4. Conclusions

Thermal inkjet printing has found several applications in the fields of tissue engineering and pharmaceutics as it is a versatile technology. There are several remarkable studies in which TIJP has been used for the development of cell-laden bioinks with excellent viability and tissue regeneration. This is not always the case, and therefore, print process optimization is a prerequisite, including understanding the effect on the cell viability and proliferation. Nevertheless, TIJP can be used to investigate new materials or combinations thereof with unique properties such as biocompatibility and high print resolution. A major advantage of the technology is the precise and accurate printing of cells in comparison with cell seeding.

Furthermore, it is a very promising technology that can lead to the commercialization of various pharmaceutical products, especially for personalized dosage forms at the point of care. However, there are several considerations that need to be taken into account. TIJP is not very easy to operate and requires significant expertise by the operator including a good understanding of troubleshooting. Specific attention should be given in the development of printable inks and the selection of the drug carriers. For the first, the choice of liquid ink is important for ensuring complete drug solubilization and fast drying times, which quite often limit the applicability of the technology: Large doses with long dry times will increase production times and thus limit the manufacturability of the technology. The selection of the polymers is equally important as they should ensure the stability of the embedded drug in the matrix, prevent recrystallization and definitely improve or tune the dissolution rates of the drug substances. Additionally, it is clear that the regulatory environment is not yet developed for entirely flexible dosage and patient-adaptable multidrug drugs, and this means that these components of printed drug delivery devices will require careful attention and experience difficulties. Quality control is a crucial component that has not

been addressed yet; there are no studies in this direction. In our opinion, TIJP could be applicable for the printing of potent APIs in small doses and particularly for the printing of QR codes. Nevertheless, there is still much work to go before we witness the full exploitation of TIJP at commercial scale or at the point of care such as in clinical pharmacies.

**Author Contributions:** Conceptualization, M.J.U. and D.D.; resources, M.J.U. and D.D.; writing—original draft preparation, M.J.U., J.H. and D.D.; writing—review and editing, M.J.U., J.H. and D.D.; supervision, M.J.U. and D.D.; project administration, M.J.U. and D.D.; funding acquisition, M.J.U. and D.D. All authors have read and agreed to the published version of the manuscript.

**Funding:** This research received no external funding.

**Institutional Review Board Statement:** Not applicable.

**Informed Consent Statement:** Not applicable.

**Data Availability Statement:** There are no additional raw data for this paper. The paper only uses secondary data from published papers, and all credits for these data have been made via citations and copyright permissions.

**Conflicts of Interest:** The authors declare no conflict of interest.

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
