# Peer review of "Thermal Inkjet Printing: Prospects and Applications in the Development of Medicine"

_technologies, doi:10.3390/technologies10050108_

Round 1

Reviewer 1 Report

1.     This review paper covers the introduction, design and application of thermal inkjet printing in the biomedical field. Given the extremely wildly used thermal inkjet printing in our daily life, I think this review paper will be helpful for biomedical technology after significant improvement is made.

2.     I can see a lot of work was done to gather the relevant information in the paper, but the information somehow scatters and need to be further organized. The principle of inkjet printing is not clear and some of them are misleading. There are many illustrations for explaining the inkjet printing technique but “zero” illustration for the published applications. Some figure captions and result explanation are too simplified. Below, some comments and suggestions are detailed for authors.

3.     Figure 1 is not an ideal illustration for a reader who is new to inkjet printing. Nozzle shouldn’t be a straight channel. Besides, the arrow actually points a nozzle channel, not a nozzle. Please see if there is a better alternative.

4.     Line 61-64, the principle of drop ejection does not sound chronologically correct. Please confirm the order of each step in other literature, such as [J. Alamán, R. Alicante, J.I. Peña, C. Sánchez-Somolinos, Inkjet printing of functional materials for optical and photonic applications, Materials (Basel). 9 (2016). https://doi.org/10.3390/ma9110910]. It should be in the order of voltage, heat/ or deformation (depending on the type of printhead), channel volume change, pressure wave and the drop ejection. It is better to explain the drop ejection with the aid of a good graph.

5.     In Figure 2, please describe what system is used in this figure. What molecule is undergoing self-assembly? For what purpose this inkjet printing is?

6.     Figure 3, please describe at what condition will lead to (a), (b) and (c)? For example, strong interaction, weak interaction etc.

7.     I don’t think Table 1 is informative when there is no actual number or range to be compared. Authors can think about removing it and focusing on the applications in the later part of this paper.

8.     Line 110-112, please confirm the jetting behaviors at different Z again. The explanation is incorrect.

9.     Figure 4 is better to be demonstrated along with pictures of the final pattern deposited by a single drop or satellite drops. Ink fluid used should be always mentioned in the figure caption.

10.  Table 2. I recommend a different way to present the ratio of two fluids, such as ethylene glycol/water (50/50). Viscosity and surface tension of water at different temperature are well documented. Authors can find the values and refer to a literature in the footnote. Don’t leave them blank.

Author mentioned the fluid properties dependent jettability in this section. In this table, it is thereby important to provide the jetting performance of those fluid if available. Are they forming single drop, satellite drop or splashing while jetting? The concentration and solvent of sodium alginate should be indicated in the SN #2. I don’t see any point the authors would like to keep (G) and (W) in SN #5. Similar problem in the later text, many abbreviations are shown for only one time.

11.  I suggest the authors try to simplify section 1.3 as many details are not addressed. For example, authors didn’t talk about what the “major drawbacks” of other inkjet printing technology are in line 153. Illustration in Figure 6 has many information but is not well explained. Section 1.3 can just focus on the difference between thermal and piezoelectric printheads before bringing out the main role of this paper, thermal inkjet printing.

12.  Line 166-168, there do exist isolated piezoelectric printheads which can be separated from the printer and sterilized, such as the ones from Dimatix printheads.

13.  Is Figure 9(b) part of Figure 9(a)? Can author indicate where the transducers are in Fig.9(b)?

14.  The caption in Figure 12 is too simple. Is Figure 12 using a printhead with single printhead? It looks like a needle, not a nozzle.

15.  Line 237, jetting with 2 µs as pulse width. Is it close to the limit of TIJP (line 188)? Any adverse effect on the jetting behavior?

16.  Line 256, what is the two-step process? Please briefly describe it.

17.  Line 283, What is “the delivery of the printed substrates”? Is it supposed to be “the delivery of sth from the printed substrates”?

18.  As the title of paper is “thermal” inkjet printing for medicine, everyone cares the thermal degradation of active compounds and cell viability. Authors did talk about this issue in the text but didn’t organize it in the table. I would suggest, table 3 is the best place to add this information. It will be very helpful to add one more column about “noticeable thermal damage and its solution”.

19.  Table 3 can be organized according to the final product is 2D flat pattern or 3D object. The concentration and solvent used of bioink should be always mentioned, much better with viscosity and surface tension of bioink if available.

20.  Why some literature in the text is not found in Table 3 such as [102] and [103]?

21.  Authors need to demonstrate 1-2 good illustration or experimental results from the literature for section 3.1.

22.  Authors need to find a better position to insert Line 294-297. It doesn’t flow with the previous sentences.

23.  Authors don’t need the sub-level of 3.2.1

24.  Authors may want to acknowledge “binder jet” in Figure 13. Binder jet is a very important inkjet-based 3D printing for making oral dosage form. Especially, there is already FDA approved 3D printed tablets using binder jet. To get more information, please refer to this paper where the thermal inkjet printhead was also used. [S.Y. Chang, S.W. Li, K. Kowsari, A. Shetty, L. Sorrells, K. Sen, K. Nagapudi, B. Chaudhuri, A.W.K. Ma, Binder-Jet 3D Printing of Indomethacin-laden Pharmaceutical Dosage Forms, J. Pharm. Sci. 109 (2020) 3054–3063. https://doi.org/10.1016/j.xphs.2020.06.027.]

25.  In Figure 14, authors may want to briefly talk about the major difference of two printers in the figure caption?

26.  Authors need to create a table like Table 3 for section 3.2 and also demonstrate 1-2 good illustration or experimental results from the literature. It would be better to illustrate TIZ-SFD which helps clarify the process that was mentioned in the text.

Author Response

#Reviewer 1 Round 1:

  1. This review paper covers the introduction, design and application of thermal inkjet printing in the biomedical field. Given the extremely wildly used thermal inkjet printing in our daily life, I think this review paper will be helpful for biomedical technology after significant improvement is made.

Answer to reviewer:

Authors would like to thank the reviewer for the constructive comments.

  1. I can see a lot of work was done to gather the relevant information in the paper, but the information somehow scatters and need to be further organized. The principle of inkjet printing is not clear and some of them are misleading. There are many illustrations for explaining the inkjet printing technique but “zero” illustration for the published applications. Some figure captions and result explanation are too simplified. Below, some comments and suggestions are detailed for authors.

Answer to reviewer:

The reviewer is right and we have now added more cases from published work including images.

  1. Figure 1 is not an ideal illustration for a reader who is new to inkjet printing. Nozzle shouldn’t be a straight channel. Besides, the arrow actually points a nozzle channel, not a nozzle. Please see if there is a better alternative.

Answer to reviewer:

The reviewer is right and we have replaced the image

  1. Line 61-64, the principle of drop ejection does not sound chronologically correct. Please confirm the order of each step in other literature, such as [J. Alamán, R. Alicante, J.I. Peña, C. Sánchez-Somolinos, Inkjet printing of functional materials for optical and photonic applications, Materials (Basel). 9 (2016). https://doi.org/10.3390/ma9110910]. It should be in the order of voltage, heat/ or deformation (depending on the type of printhead), channel volume change, pressure wave and the drop ejection. It is better to explain the drop ejection with the aid of a good graph.

Answer to reviewer:

Droplet formation in line no 61-64 (line 62-65 in updated version of manuscript) is a piezoelectric process, taken from ref. 37 (Singh, M.; Haverinen, H.M.; Dhagat, P.; Jabbour, G.E. Inkjet Printing—Process and Its Applications. Adv. Mater. 2010, 22, 673–685, doi:10.1002/ADMA.200901141.). The ref. reviewer has mentioned has described CIJ, thermal and piezoelectric together, which has resemblance with our writings in line number 230-235, where thermal inkjet printing process has been described.

  1. In Figure 2, please describe what system is used in this figure. What molecule is undergoing self-assembly? For what purpose this inkjet printing is?

Answer to reviewer:

We have added more details in the figure title which explains the actual schematic representation. It is figure 1 right now in the updated version of our manuscript.

  1. Figure 3, please describe at what condition will lead to (a), (b) and (c)? For example, strong interaction, weak interaction etc.

Answer to reviewer:

We have now removed Figure 3 in order to add more figures from actual case studies.

  1. I don’t think Table 1 is informative when there is no actual number or range to be compared. Authors can think about removing it and focusing on the applications in the later part of this paper.

Answer to reviewer:

We have kept Table 1 as it will be useful for the readers as it provides a comparison with other printing technologies.

  1. Line 110-112, please confirm the jetting behaviors at different Z again. The explanation is incorrect.

Answer to reviewer:

We have cross-checked the text with the relative publications and the Z ranges are correct.

  1. Figure 4 is better to be demonstrated along with pictures of the final pattern deposited by a single drop or satellite drops. Ink fluid used should be always mentioned in the figure caption.

Answer to reviewer:

We have now added the ink fluid in the Figure details.

  1. Table 2. I recommend a different way to present the ratio of two fluids, such as ethylene glycol/water (50/50). Viscosity and surface tension of water at different temperature are well documented. Authors can find the values and refer to a literature in the footnote. Don’t leave them blank.

Answer to reviewer:

We followed the reviewer’s comment and adjusted Table 2. Total of 8 ratios have been added for ethylene glycol/water with the value of viscosity and surface tension.

- Author mentioned the fluid properties dependent jettability in this section. In this table, it is thereby important to provide the jetting performance of those fluid if available. Are they forming single drop, satellite drop or splashing while jetting? The concentration and solvent of sodium alginate should be indicated in the SN #2. I don’t see any point the authors would like to keep (G) and (W) in SN #5. Similar problem in the later text, many abbreviations are shown for only one time.

Answer to reviewer:

Table 2 has been updated as per reviewer’s comments. However, it was not possible to find the Z value for all the inks that’s why authors decided not to add it, and none of the experimental studies in this table mentioned of forming satellite droplet while jetting. The reason could be, since the values has been optimized until those experiments have been successful.

  1. I suggest the authors try to simplify section 1.3 as many details are not addressed. For example, authors didn’t talk about what the “major drawbacks” of other inkjet printing technology are in line 153. Illustration in Figure 6 has many information but is not well explained. Section 1.3 can just focus on the difference between thermal and piezoelectric printheads before bringing out the main role of this paper, thermal inkjet printing.

Answer to reviewer:

Authors completely agree with the reviewer and the section 1.3 has been updated. Authors have talked about the drawbacks of inkjet printing. Fig. 6 has been adjusted accordingly.

  1. Line 166-168, there do exist isolated piezoelectric printheads which can be separated from the printer and sterilized, such as the ones from Dimatix printheads.

Answer to reviewer:

We were not familiar with this information and we have now modified the sentence.

  1. Is Figure 9(b) part of Figure 9(a)? Can author indicate where the transducers are in Fig.9(b)?

Answer to reviewer:

We have removed the figure 9

  1. The caption in Figure 12 is too simple. Is Figure 12 using a printhead with single printhead? It looks like a needle, not a nozzle.

Answer to reviewer:

Authors would like to thank the reviewer for helping us making the better version of this manuscript. Figure 12 has been removed.

  1. Line 237, jetting with 2 µs as pulse width. Is it close to the limit of TIJP (line 188)? Any adverse effect on the jetting behavior?

Answer to reviewer:

Line 188 (mentions the generalized optimum range and line 237 presents the optimized value used in that respective experiment (ref. no 105) which is close to the limit mentioned in line 188. No adverse effect on the jetting behavior has been mentioned by the authors of respective article.

  1. Line 256, what is the two-step process? Please briefly describe it.

Answer to reviewer:

We have added the two steps in the text and described them. Reviewer can find it in line number 292-305.

  1. Line 283, What is “the delivery of the printed substrates”? Is it supposed to be “the delivery of sth from the printed substrates”?

Answer to reviewer:

The reviewer is right as the correct word was “deposition" and not delivery.  We have now made the changes. Reviewer can find it corrected in line 375 in the updated version of our manuscript.

  1. As the title of paper is “thermal” inkjet printing for medicine, everyone cares the thermal degradation of active compounds and cell viability. Authors did talk about this issue in the text but didn’t organize it in the table. I would suggest, table 3 is the best place to add this information. It will be very helpful to add one more column about “noticeable thermal damage and its solution”.

Answer to reviewer:

Authors have mentioned the temperature issue (find in line 315-319) but did not discussed it elaborately because all published experiments (for example ref 105) mention the optimized value for temperature which did not influence any alarming percentage of cell damages. We have mentioned both the positive and negative outcomes in table 3. In addition, when a value like 80% cell viability is mentioned, it automatically indicates the 20% cell damage.

  1. Table 3 can be organized according to the final product is 2D flat pattern or 3D object. The concentration and solvent used of bioink should be always mentioned, much better with viscosity and surface tension of bioink if available.

Answer to reviewer:

Authors appreciate reviewer’s idea however; it will be too much information to add in the Table.  We believe that the readers can use the Table and extract additional information from the published article.

  1. Why some literature in the text is not found in Table 3 such as [102] and [103]?

Answer to reviewer:

Author would like to thank the reviewer for noticing the error and help us making the better version of our manuscript. We have added the mentioned references in the table 3. Reviewer can find it in row 19 & 20.

  1. Authors need to demonstrate 1-2 good illustration or experimental results from the literature for section 3.1.

Answer to reviewer:

Authors genuinely appreciate the reviewer’s idea and updated the manuscript.

  1. Authors need to find a better position to insert Line 294-297. It doesn’t flow with the previous sentences.

Answer to reviewer:

The reviewer is right and we have removed the lines.

  1. Authors don’t need the sub-level of 3.2.1

Answer to reviewer:

Authors would like to thank the reviewer for helping us making the better version of this manuscript by noticing the mistake and it has been corrected.

  1. Authors may want to acknowledge “binder jet” in Figure 13. Binder jet is a very important inkjet-based 3D printing for making oral dosage form. Especially, there is already FDA approved 3D printed tablets using binder jet. To get more information, please refer to this paper where the thermal inkjet printhead was also used.[S.Y. Chang, S.W. Li, K. Kowsari, A. Shetty, L. Sorrells, K. Sen, K. Nagapudi, B. Chaudhuri, A.W.K. Ma, Binder-Jet 3D Printing of Indomethacin-laden Pharmaceutical Dosage Forms, J. Pharm. Sci. 109 (2020) 3054–3063. https://doi.org/10.1016/j.xphs.2020.06.027.]

Answer to reviewer:

Thanks to the reviewer for helping us making the better version of manuscript. We have updated the subheading 3.2 with the information suggested by the reviewer. Reviewer can find it in line 397-403.

  1. In Figure 14, authors may want to briefly talk about the major difference of two printers in the figure caption?

Answer to reviewer:

Figure 14 has been removed.

  1. Authors need to create a table like Table 3 for section 3.2 and also demonstrate 1-2 good illustration or experimental results from the literature. It would be better to illustrate TIZ-SFD which helps clarify the process that was mentioned in the text.

Answer to reviewer:

Authors appreciate the reviewer’s idea and added a table (kindly see table 3) and figure in section 3.2.

Reviewer 2 Report

Good, timely and comprehensive review of thermal inkjet printing and bioprinting. 

Author Response

#Reviewer 2 Round 1:

Good, timely and comprehensive review of thermal inkjet printing and bioprinting.

Answer to reviewer:

Authors would like to thank the reviewer for the appreciative comment

Reviewer 3 Report

I really enjoyed reading this paper. I believe it sets out the current position of InkJet 3DP very well. Certainly if you are new to this field it is a great review and easy to follow.

A few thoughts came to mind while reading the paper, but these are primarily expansions to the paper as opposed to corrections. However they would make it more complete if included:

- the ability to create shapes and produce modified release hasn't been touched upon. This topic is important as I know that 3DP has the ability to create some very sophisticated release profiles. Is this possible with InkJet or is this one of the limitations?

- children and elderly are quoted as possibly one of main benefactors of 3DP and personalised medicines (I agree), but you discuss excipients such as alcohol and propylene glycol with no reference to whether they may be an issue - are the amounts below recommended consumption (or lower than in normal tablets)

Specific:

Line 322 mentions PARACETAMOL and line 326 mentions ACETAMINOPHEN - these are the same drug and should probably be stated as the same 

Author Response

#Reviewer 3 Round 1:

  1. I really enjoyed reading this paper. I believe it sets out the current position of InkJet 3DP very well. Certainly if you are new to this field it is a great review and easy to follow.

Answer to reviewer:

Authors would like to thank the reviewer for the appreciative comment.

  1. A few thoughts came to mind while reading the paper, but these are primarily expansions to the paper as opposed to corrections. However, they would make it more complete if included:

- the ability to create shapes and produce modified release hasn't been touched upon. This topic is important as I know that 3DP has the ability to create some very sophisticated release profiles. Is this possible with InkJet or is this one of the limitations?

Answer to reviewer:

We have added relevant discussion of how to make multilayered structures.  Indeed the technology can be used for complex geometries and customized release profiles with the selection of appropriate polymer grades.

- children and elderly are quoted as possibly one of main benefactors of 3DP and personalized medicines (I agree), but you discuss excipients such as alcohol and propylene glycol with no reference to whether they may be an issue - are the amounts below recommended consumption (or lower than in normal tablets)

Answer to reviewer:

We agree with the reviewer but the discussion of pediatric excipients is not align to the aim of the review. This has been addressed by other colleagues including a wide range of dosage forms

Specific:

Line 322 mentions PARACETAMOL and line 326 mentions ACETAMINOPHEN - these are the same drug and should probably be stated as the same

Answer to reviewer:

Authors completely agree with the reviewer that these are same drug however, the lines reviewer has mentioned (416 & 419 in the updated version of our manuscript) are from two different case studies (ref 142 & 143). Authors have used the same terms used by the authors of original research paper ( ref 142 & 143) out of respect for their published experimental works.

Reviewer 4 Report

 Comments:

1.      The authors should first discuss the different bioprinting techniques and highlight the advantages and limitations of inkjet bioprinting over other bioprinting techniques according to ASTM standards (material jetting, material extrusion and vat polymerization bioprinting) with relevant references. The authors can refer to some recent papers below.

a.      Material extrusion

                                                    i.     "Extrusion bioprinting of soft materials: An emerging technique for biological model fabrication." Applied Physics Reviews 6, no. 1 (2019): 011310.

b.      Material jetting

                                                    i.     "Inkjet bioprinting of biomaterials." Chemical Reviews 120, no. 19 (2020): 10793-10833.

c.      Vat polymerization

                                                    i.     "Vat polymerization-based bioprinting—process, materials, applications and regulatory challenges." Biofabrication 12, no. 2 (2020): 022001.  

2.      The authors should elaborate more on the droplet drying process using the different evaporation modes (constant contact radius, stick-slide or mixed evaporate mode) in Section 1.1.

3.      It is recommended that the authors also elaborate more on how droplet evaporation would potentially affect the cell viability in inkjet bioprinting?

4.      The authors should discuss more in-depth on the different types of inks (Section 1.2.) for different applications (e.g tissue engineering, biomedical applications) and include additional column (applications) for Table 2.

5.      Most of the information discussed in Section 1.2. are easily available from most review papers.

a.      Is there any recent progress in ink development for thermal inkjet printing to improve the printability or accuracy of different inks?

b.      What are some of the printing strategies to get multi-layered structures using inkjet printing??

6.      Do check if copyrights permission has been obtained for Figure 6B, also use a figure of higher resolution.  

7.      It is recommended that the authors elaborate on the effect of different inkjet printing technology on the viability of printed cells. What is the effect of high temperature on printed cells?

8.      As there is a growing trend of using deep learning/machine learning in 3D printing, the authors should include a section on the using deep learning/machine learning in inkjet printing.

a.      "Deep learning for fabrication and maturation of 3D bioprinted tissues and organs." Virtual and Physical Prototyping 15, no. 3 (2020): 340-358.

b.      "Unsupervised learning for the droplet evolution prediction and process dynamics understanding in inkjet printing." Additive Manufacturing 35 (2020): 101197.

Author Response

#Reviewer 4 Round 1:

  1. The authors should first discuss the different bioprinting techniques and highlight the advantages and limitations of inkjet bioprinting over other bioprinting techniques according to ASTM standards (material jetting, material extrusion and vat polymerization bioprinting) with relevant references. The authors can refer to some recent papers below.
  2. Material extrusion
  3. "Extrusion bioprinting of soft materials: An emerging technique for biological model fabrication." Applied Physics Reviews 6, no. 1 (2019): 011310.
  4. Material jetting
  5. "Inkjet bioprinting of biomaterials." Chemical Reviews 120, no. 19 (2020): 10793-10833.
  6. Vat polymerization
  7. "Vat polymerization-based bioprinting—process, materials, applications and regulatory challenges." Biofabrication 12, no. 2 (2020): 022001.  

Answer to reviewer:

We have added the relevant articles as the reviewer suggested (line no 295-301). However, the aim of the article is not to go in depth about the differences between various printing technologies.  We have add a brief overview in Table 1 to give a general idea to the readers.

  1. The authors should elaborate more on the droplet drying process using the different evaporation modes (constant contact radius, stick-slide or mixed evaporate mode) in Section 1.1.

 Answer to reviewer:

Authors appreciate reviewer’s suggestion and elaborated the droplet drying process.

  1. It is recommended that the authors also elaborate more on how droplet evaporation would potentially affect the cell viability in inkjet bioprinting?

 Answer to reviewer:

The relative discussion has been added.

  1. The authors should discuss more in-depth on the different types of inks (Section 1.2.) for different applications (e.g tissue engineering, biomedical applications) and include additional column (applications) for Table 2.

Answer to reviewer:

We agree with the reviewer that there are many different inks that can be designed for various applications. We have updated the subheading 1.2. However, the reported studies of inks used by TIJP are limited and do not cover as many applications’ and other printing technologies (e.g., extrusion based)

  1. Most of the information discussed in Section 1.2. are easily available from most review papers.
  2. Is there any recent progress in ink development for thermal inkjet printing to improve the printability or accuracy of different inks?

Answer to reviewer:

This paper focuses on the TIJP in terms of its uses in development of medicine and has already discussed a good amount of recently published case studies regarding it which mentions the bioinks used in that respective experimental works. We have added some more newer cases in the article now.

  1. What are some of the printing strategies to get multi-layered structures using inkjet printing??

 Answer to reviewer:

Authors have added strategies such as Photolithography, Micro-contact printing, Shadow mask. Reviewer can find it in Table 1. Also, strategies like- Screen printing, low-cost inkjet piezoelectric printers etc. have now been added to the manuscript.

  1. Do check if copyrights permission has been obtained for Figure 6B, also use a figure of higher resolution.  

Answer to reviewer:

Authors would like to inform the reviewer that copyright permissions for every single figure used in this manuscript has already been taken.

  1. It is recommended that the authors elaborate on the effect of different inkjet printing technology on the viability of printed cells. What is the effect of high temperature on printed cells?

 Answer to reviewer:

Authors appreciate the recommendation of reviewer and updated the manuscript.

Round 2

Reviewer 1 Report

Draft content has been significantly improved while there is still critical information that need to be modified, or confirmed, or addressed. By the way, many line numbers that authors provided in the responses are not up to date. I had to search for them by comparing the old and new version. Please do check the line number in the responses again after the final revision is done.

Figure 2 caption. It is not the inkjet printing mechanism. It just represents two type of inkjet printing systems. Please correct it.

Line 62-66, the description still doesn’t sound correct for a general drop ejection process in inkjet. Authors need to draw a better description of how a drop is generated for either thermal or piezo printhead. I have checked the paper that authors provided (Singh, M.; Haverinen, H.M.; Dhagat, P.; Jabbour, G.E. Inkjet Printing—Process and Its Applications. Adv. Mater. 2010, 22, 673–685, doi:10.1002/ADMA.200901141). Unfortunately, the process they mentioned is solely for a piezo printhead. Why do a thermal jet printhead need a piezo action? Actually, many good principle information can be found in printhead manufacturer websites, such as (https://www.imaging.org/site/IST/Resources/Imaging_Tutorials/How_an_Ink_Jet_Printer_Works/IST/Resources/Tutorials/Inkjet_Printer.aspx?hkey=5c0e9b54-b357-4dbb-b440-f07557f5163e). After authors read and understand it, please still find a good reference paper to support your sentence in the draft.

Line 65, deformation of nozzle chamber was driven by either piezo deformation or bubble generation. There is no way that chamber deformation leads to an external voltage. That is what I meant, it is not chronologically correct.

Figure 3 caption. I don’t understand what is “6,13-bis((triisopropylsilylethynyl) pentacene (form 20% dodecane)”. What is the percentage of solute, what is solvent?

Line 174-175. Jettable region is generally considered within z=1 to z=10 as authors mentioned in line 173. Why would authors say “ if Z value is less than 1 (Z<10) then the 174 formation of satellite droplets occurs (also termed as secondary droplets)><10) then the formation of satellite droplets occurs (also termed as secondary droplets)”? Satellite droplets are usually observed as Z > 10.

Figure 5. Authors said the liquid type has been added. I didn’t see it.

Table 2. Authors said, the Z is not provided for most of inks in literature. I checked the reference 70 and 74 where most inks are taken from. Z values are well documented! Please do check the references thoroughly and add Z value if it is available.

Line 226-228, do authors happen to find any rationales for this?

Line 229-230, I would day sterilization is no longer unique for thermal inkjet printhead, many piezo printhead can do. This feature shouldn’t stand out.

Line 233-242, this paragraph can combine with line 223-231 and refer to Figure 6B which include all IJP that authors mentioned here.

Line 242, authors want to move this sentence to line 199, right after “(DOD)” and refer to Figure 6A

Line 336-356, I understand saponification could modified the viscosity of inks but I don’t catch what has changed the inks chemically after reading all of this? Can authors just explain its mechanism in brief?

Line 428, replace “metal” with “powder”. Binder-jet can work with different type of powders. For medicine, no one would use metal powder.

Figure 9. Is the 3D rotary printing equipped with a thermal inkjet printhead?

Figure 13, reference?

Table 4, do authors consider ref 142 to 147 be part of this table?

Author Response

# Round 2 Reviewer 1

  1. Figure 2 caption. It is not the inkjet printing mechanism. It just represents two type of inkjet printing systems. Please correct it.

 Answer to the reviewer:

Authors would like to thank Reviewer for the suggestion. We have reordered the figure 2 which is now figure 5.

  1. Line 62-66, the description still doesn’t sound correct for a general drop ejection process in inkjet. Authors need to draw a better description of how a drop is generated for either thermal or piezo printhead. I have checked the paper that authors provided (Singh, M.; Haverinen, H.M.; Dhagat, P.; Jabbour, G.E. Inkjet Printing—Process and Its Applications. Adv. Mater. 2010, 22, 673–685, doi:10.1002/ADMA.200901141). Unfortunately, the process they mentioned is solely for a piezo printhead. Why do a thermal jet printhead need a piezo action? Actually, many good principle information can be found in printhead manufacturer websites, such as (https://www.imaging.org/site/IST/Resources/Imaging_Tutorials/How_an_Ink_Jet_Printer_Works/IST/Resources/Tutorials/Inkjet_Printer.aspx?hkey=5c0e9b54-b357-4dbb-b440-f07557f5163e). After authors read and understand it, please still find a good reference paper to support your sentence in the draft.

Answer to the reviewer:

We have modified the and adjusted the discussion in lines 60-66 which refers to inkjet printing through the use of a piezoelectric system.  The mechanism of thermal inkjet printing is discussed in detail in lines 267-276.

  1. Line 65, deformation of nozzle chamber was driven by either piezo deformation or bubble generation. There is no way that chamber deformation leads to an external voltage. That is what I meant, it is not chronologically correct.

Answer to the reviewer:

It has been addressed in the previous query of the reviewer

  1. Figure 3 caption. I don’t understand what is “6,13-bis((triisopropylsilylethynyl) pentacene (form 20% dodecane)”. What is the percentage of solute, what is solvent?

Answer to the reviewer:

It was 1 µL 6,13- bis((triisopropylsilylethynyl) pentacene (TIPS_PEN) solution, where TIPS_PEN crystals were solute and chlorobenzene/dodecane were solvent mixture. The reviewer can find the details in page no. 231-232 of ref no. 37 (Lim, J.A.; Lee, W.H.; Lee, H.S.; Lee, J.H.; Park, Y.D.; Cho, K. Self-Organization of Ink-Jet-Printed Triisopropylsilylethynyl Pentacene via Evaporation-Induced Flows in a Drying Droplet. Adv. Funct. Mater. 2008, 18, 229–234, doi:10.1002/ADFM.200700859.). For reviewer’s information, these details have been included in the caption because another reviewer suggested us to mention it.

  1. Line 174-175. Jettable region is generally considered within z=1 to z=10 as authors mentioned in line 173. Why would authors say “ if Z value is less than 1 (Z<10) then the 174 formation of satellite droplets occurs (also termed as secondary droplets)><10) then the formation of satellite droplets occurs (also termed as secondary droplets)”? Satellite droplets are usually observed as Z > 10.

Answer to the reviewer:

Authors would like to request the reviewer look at line 178-179 rather in line 174-175 for the latest findings of researchers. The limit is 1<Z<14. Line 166-182, the authors have explained the findings of researchers.

  1. Figure 5. Authors said the liquid type has been added. I didn’t see it.

Answer to the reviewer:

We have included the liquid type for Fig 2 for the schematic diagram. However, there is no information by the authors for the liquid used in Fig. 5 (now figure 4).

  1. Table 2. Authors said, the Z is not provided for most of inks in literature. I checked the reference 70 and 74 where most inks are taken from. Z values are well documented! Please do check the references thoroughly and add Z value if it is available.

Answer to the reviewer:

Authors agree with the reviewer. We have added z values for the maximum numbers of inks mentioned in the table 2.

  1. Line 226-228, do authors happen to find any rationales for this?

Answer to the reviewer:

Yes. Authors find line 226-228 (line 229-231 in the latest version of manuscript) rationale for this since it helps to draw a difference between two separate methods.

  1. Line 229-230, I would day sterilization is no longer unique for thermal inkjet printhead, many piezo printhead can do. This feature shouldn’t stand out.

Answer to the reviewer:

Lines have been deleted as per reviewer’s suggestion.

  1. Line 233-242, this paragraph can combine with line 223-231 and refer to Figure 6B which include all IJP that authors mentioned here.

Answer to the reviewer:

Author would like to thank to the reviewer for this suggestion. however, authors would like to keep it for the better expression.

  1. Line 242, authors want to move this sentence to line 199, right after “(DOD)” and refer to Figure 6A

Answer to the reviewer:

We have reordered figure 2 and mentioned it right after “(DOD)”.

  1. Line 336-356, I understand saponification could modified the viscosity of inks but I don’t catch what has changed the inks chemically after reading all of this? Can authors just explain its mechanism in brief?

Answer to the reviewer:

The process is discussed in detail in the provided references and we would like to keep it short as it is out of the scope of the review.

  1. Line 428, replace “metal” with “powder”. Binder-jet can work with different type of powders. For medicine, no one would use metal powder.

Answer to the reviewer:

Line 428 (line 424 after using track change in the latest version of manuscript), the word metal has been replaced with powder.

  1. Figure 9. Is the 3D rotary printing equipped with a thermal inkjet printhead?

Answer to the reviewer:

No. It does not. Figure 9 has only showed that heat treatment increased the printability of ink material.

  1. Figure 13, reference?

Answer to the reviewer:

Thank you for noticing the mistake. Reference has been added.

  1. Table 4, do authors consider ref 142 to 147 be part of this table?

Answer to the reviewer:

No! none of these refs mention a TIJ printer. Table 4 only mentions the TIJ printers that have been used in fabricating oral dosage forms.

Reviewer 4 Report

The authors have addressed all the comments, the revised manuscript can be accepted in present form

Author Response

We would like to thank the reviewer for his kind acceptance.